# Learning to Sample and Aggregate: Few-shot Reasoning over Temporal Knowledge Graphs

**Ruijie Wang**[1], **Zheng Li**[2], **Dachun Sun**[1], **Shengzhong Liu**[1], **Jinning Li**[1],
**Bing Yin**[2], **Tarek Abdelzaher**[1]

[1]University of Illinois at Urbana Champaign, IL, USA
[2]Amazon.com Inc, CA, USA
{ruijiew2, dsun18, sl29, jinning4, zaher}@illinois.edu
{amzzhe, alexbyin}@amazon.com

## Abstract

In this paper, we investigate a realistic but underexplored problem, called few-shot temporal knowledge graph reasoning, that aims to predict future facts for newly emerging entities based on extremely limited observations in evolving graphs. It offers practical value in applications that need to derive instant new knowledge about new entities in temporal knowledge graphs (TKGs) with minimal supervision. The challenges mainly come from the *few-shot* and *time shift* properties of new entities. First, the limited observations associated with them are insufficient for training a model from scratch. Second, the potentially dynamic distributions from the initially observable facts to the future facts ask for explicitly modeling the evolving characteristics of new entities. We correspondingly propose a novel Meta Temporal Knowledge Graph Reasoning (`MetaTKGR`) framework. Unlike prior work that relies on rigid neighborhood aggregation schemes to enhance low-data entity representation, `MetaTKGR` dynamically adjusts the strategies of sampling and aggregating neighbors from recent facts for new entities, through temporally supervised signals on future facts as instant feedback. Besides, such a meta temporal reasoning procedure goes beyond existing meta-learning paradigms on static knowledge graphs that fail to handle temporal adaptation with large entity variance. We further provide a theoretical analysis and propose a temporal adaptation regularizer to stabilize the meta temporal reasoning over time. Empirically, extensive experiments on three real-world TKGs demonstrate the superiority of `MetaTKGR` over state-of-the-art baselines by a large margin.

## 1 Introduction

Temporal Knowledge Graphs (TKGs) [34, 3, 22, 55] store temporally evolving facts (By fact, we refer to a subject entity has a relation with an object entity at some time, e.g., "*Macron*" was "*elected*" as "*French president*" in "*2017*"). It has been increasingly used in various knowledge-driven and time-sensitive applications, such as social event forecasting [38, 24, 53], question answering [40], and recommendation [70, 69, 54]. Large scale TKGs are hard to obtain due to the time-varying nature of labels and the excessive cost of human annotation. As such, automating prediction and reasoning about missing facts over time has attracted more recent interest [47, 10, 13, 48, 20]. However, most prior efforts are limited to reasoning about existing entities but neglecting the commonly observed low-data regime where many new entities emerge with extremely few observations as new events or topics evolve over time [44]. This restricts their applicability.

Instead, inspired by the human ability to recognize new concepts from exposure to only very few instances, we propose and solve a practical and challenging *few-shot temporal knowledge graph*

36th Conference on Neural Information Processing Systems (NeurIPS 2022).

*reasoning* problem that specifically aims to predict future facts for **newly emerging entities** with a few observed links on TKGs. Existing efforts [15, 52, 1, 8] in similar settings simplify the task by ignoring the emergence of new entities over time and randomly simulating unseen entities on static knowledge graphs. This formulation, originating from few-shot learning in the vision domain [11, 39], fails at capturing the real-world divergence between the distributions of seen and unseen entities. To the best of our knowledge, the generalized few-shot temporal reasoning task, that aims to imitate the fast learning ability of humans, has still not been formally investigated.

The challenges in solving this problem are two-fold: 1) **few-shot**: the extremely few facts associated with new entities cannot provide sufficient information for representation learning from scratch. 2) **time shift**: new entities usually exhibit different characteristics in the future, due to the time-evolving nature of TKGs. It leads models to generalize poorly in the future. Unfortunately, prior work fails to overcome these two challenges simultaneously.

On one hand, to enhance low-data entity characterization, a diverse range of neighborhood aggregation schemes that retrieve related information from other entities are explored, such as hard subgraph sampling methods (e.g., vanilla hop-based sampling [41], random sampling [16], personalized Pagerank sampling [66, 54], importance sampling [6], etc) and soft sampling via trainable attention mechanisms [47, 48, 20]. However, those rigid methods cannot well adapt to the new entities with the time-varying distributions due to the lack of instant supervision. On the other hand, recent attempts [1, 68, 18, 4] leverage meta-learning to improve unseen entity adaptation. While the setup for unseen entities is simulated by randomly splitting entity sets from static graphs, they fail to handle the more realistic temporal adaptation with large entity variances.

To overcome both the aforementioned challenges, we propose a novel Meta Temporal Knowledge Graph Reasoning (`MetaTKGR`) framework, where learning the strategies for sampling and aggregating neighbors from recent facts (to enhance the new entities' predictions) can be dynamically adapted from signals of temporal supervision on their future facts as instant feedback. Intuitively, the global knowledge of sampling and aggregating can be extracted as learning to learn ability through such a meta-optimization, which takes the time-evolving nature of knowledge into consideration, avoiding progressive performance drift over time. Concretely, we formulate this learning strategy to be parameterized by a temporal encoder. On the simulated new entities during training phase, starting from the initial few-shot links, the temporal encoder can softly sample and attentively integrate relative and time-aware information from the temporal neighbors, making the strategy learning (neighbor sampling + aggregation) differentiable. The quantitative measurement of how well the current strategy performs can be reflected in the performance on predicting future facts allowing one to automatically adjust the strategy in turn. During the nested loop for meta-optimization, we firstly learn the temporal encoder on recent facts (*inner loop*), then gradually increase the difficulties by autoregressively adapting it to farther away future facts (*outer loop*). Furthermore, to better estimate the supervision signal given by future facts in different and unseen distributions, we adopt the PAC-Bayes method [35] to theoretically analyze the temporal adaptation bound on the future facts. This can serve as an adaptation regularizer to provide stability and improve the generalization ability of current strategy over time. Empirically, extensive experiments on three real-world TKGs validate the effectiveness of `MetaTKGR`, which significantly outperforms all baselines from static/temporal KG reasoning and few-shot (knowledge) graph learning areas, by up to $11.4\%$ relative gain on average with competitive efficiency. To sum up, our main contributions are three-fold:

- **Problem formulation:** We explore a practical *few-shot temporal knowledge graph reasoning* setting, minimizing the gap with the realistic few-shot learning scenarios for humans;
- **Novel framework:** We propose a novel meta temporal reasoning framework on TKGs, namely `MetaTKGR`, to address both few-shot and time shift challenges;
- **Extensive evaluation:** the proposed `MetaTKGR` method demonstrates significant gains in effectiveness on three real-world TKGs over a diverse range of state-of-the-art baselines.

## 2 Problem Definition

In this section, we formally define the few-shot temporal knowledge graph reasoning task. First of all, a temporal knowledge graph can be defined as follows:

**Definition 2.1** (**Temporal Knowledge Graph**). *A temporal knowledge graph can be denoted as* $\mathcal{G}^T = \{(e_s, r, e_o, t)\} \subseteq \mathcal{E}^T \times \mathcal{R} \times \mathcal{E}^T \times \mathcal{T}$, *where* $\mathcal{E}^T$ *denotes a set of entities that appear in time*

interval $(0, T)$, $\mathcal{R}$ denotes relation set, and $\mathcal{T}$ denotes timestamp set. Each temporal link $(e_s, r, e_o, t)$ refers to the fact that a subject entity $e_s \in \mathcal{E}^T$ has a relation $r \in \mathcal{R}$ with an object entity $e_o \in \mathcal{E}^T$ at timestamp $t \in \mathcal{T}$.

As a temporal knowledge graph evolves, new entities continuously emerge, leading to an expansion of the entities set $\mathcal{E}^T$ over time. We further define new entities as follows:

**Definition 2.2 (New Entities in a Temporal Knowledge Graph).** *Given a time interval $(T, T')$ $(T' > T)$, entities that join the graph during $(T, T')$, i.e., $\tilde{e} \in \mathcal{E}^{T'} \setminus \mathcal{E}^T$ ($\setminus$ denotes setminus), are defined as new entities on time interval $(T, T')$.*

Knowledge graph reasoning (KGR) is essentially the problem of predicting missing facts in the partially observed KG. While existing few-shot KGR models simulate unseen entities by randomly selecting from the existing entities, we focus specifically on predictions for newly emerging entities over time, which can be formalized as *few-shot temporal knowledge graph reasoning* task:

**Definition 2.3 (Few-shot Temporal Knowledge Graph Reasoning).** *Given a temporal knowledge graph $\mathcal{G}^T$ collected no later than $T$, for each new entity $\tilde{e} \in \mathcal{E}^{T'} \setminus \mathcal{E}^T (T' > T)$, we assume the first $K$ associated facts are observed: $\{(\tilde{e}, r_i, e_i, t_i) \text{ or } (e_i, r_i, \tilde{e}, t_i)\}_{i=1}^{K}$. The task aims to predict the missing entities in the future facts $(\tilde{e}, r, ?, t)$ or $(?, r, \tilde{e}, t)$ given the relation $r \in \mathcal{R}$ and specific time $t \in (T, T')$. We further assume $K$ is a small number, as a realistic setting.*

## 3 Learning to Sample And Aggregate with `MetaTKGR`

In this section, we present a novel framework `MetaTKGR` to solve the few-shot temporal knowledge graph reasoning problem. We first introduce the basic setup of our learning objective and the overall framework, and then detail the temporal encoder module to represent new entities, followed by introduction of meta temporal reasoning for model training.

### 3.1 Learning Objective

Suppose we are given a temporal knowledge graph $\mathcal{G}^T = \{(e_s, r, e_o, t)\} \subseteq \mathcal{E}^T \times \mathcal{R} \times \mathcal{E}^T \times \mathcal{T}$ collected no later than time $T$. For each new entity $\tilde{e}$ appearing within a later interval $(T, T')$, we aim to predict future links $(\tilde{e}, r, e, t)$ or $(e, r, \tilde{e}, t)$ happening at timestamp $t \in (T, T')$. Towards this goal, we measure correctness of each possible quadruple by a score function $s(\cdot; \phi)$ parameterized by $\phi$, and maximize the scores of true quadruples containing any new entities in order to rank them higher than all other false quadruples:

$$\max_{\phi} \mathbb{E}_{\tilde{e} \sim p(\tilde{\mathcal{E}})}[s(\tilde{e}, r, e, t; \phi) \text{ or } s(e, r, \tilde{e}, t; \phi)], \text{ where } \tilde{\mathcal{E}} = \mathcal{E}^{T'} \setminus \mathcal{E}^T, \ t \in (T, T'), \quad (1)$$

where $p(\tilde{\mathcal{E}})$ denotes distribution of all new entities, $\phi$ denotes parameters of model to represent entity/relation into $d$-dimensional space $\mathbf{h}_{\tilde{e}}, \mathbf{h}_e, \mathbf{h}_r \in \mathbb{R}^d$. To represent $\tilde{e}$ into $\mathbf{h}_{\tilde{e}}$ via $\phi$, as aforementioned, the challenges lie in how to enable $\phi$ to encode generalized knowledge (how to sample and aggregate temporal neighbors) that can be easily adapted to new $\tilde{e}$ and achieve robust performance over time, even for a long time interval $(T, T')$. We thereby formulate few-shot temporal knowledge graph reasoning as a meta-learning problem to extract such knowledge.

### 3.2 MetaTKGR Framework

Figure 1 shows the `MetaTKGR` framework. To be more formal, each task corresponds to each new entity $\tilde{e}$ over distribution $p(\tilde{\mathcal{E}})$, and the links can be represented as a chronological sequence $\{(\tilde{e}, r_i, e_i, t_i) \text{ or } (e_i, r_i, \tilde{e}, t_i) | t_i \in (T, T'), t_i \leq t_j \text{ if } i < j\}_{i=1}^{N_{\tilde{e}}}$, where $N_{\tilde{e}}$ denotes total number of links associated to $\tilde{e}$. Then we divide them into support set $\mathcal{S}_{\tilde{e}} = \{(\tilde{e}, r_i, e_i, t_i) \text{ or } (e_i, r_i, \tilde{e}, t_i)\}_{i=1}^{K}$, and query set $\mathcal{Q}_{\tilde{e}} = \{(\tilde{e}, r_i, e_i, t_i) \text{ or } (e_i, r_i, \tilde{e}, t_i)\}_{i=K+1}^{N_{\tilde{e}}}$, where $K$ denotes the amount of initially observable facts of $\tilde{e}$.

During the meta-training phase, we simulate a set of new entities from existing entity set by assuming there are only few-shot links of these entities (support set $\mathcal{S}_{\tilde{e}}$). The model first fine-tunes parameters $\phi$ on $\mathcal{S}_{\tilde{e}}$ (*inner loop*), and then minimizes the predictive loss on corresponding query set $\mathcal{Q}_{\tilde{e}}$ using the updated parameters $\phi_{\tilde{e}}$ (*outer loop*). It optimizes the global sampling and aggregation strategy as

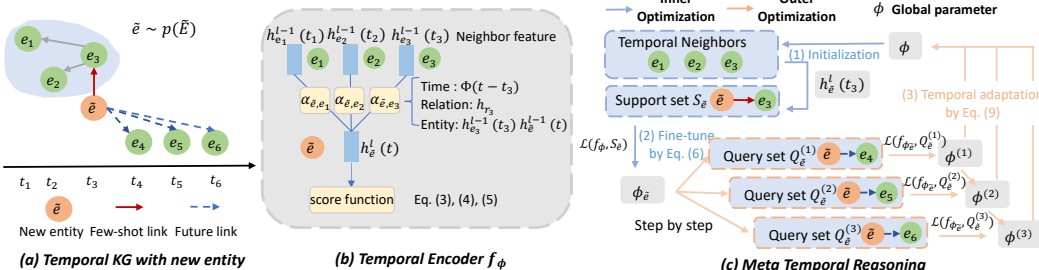

**(a) Temporal KG with new entity**   **(b) Temporal Encoder $f_\phi$**   **(c) Meta Temporal Reasoning**

Figure 1: An overview of `MetaTKGR` framework. (a) Task illustration; (b) Temporal encoder parameterized by $\phi$ samples and aggregates information from temporal neighbors; (c) Meta temporal reasoning adopts a bi-level optimization to learn $\phi$. In the **inner optimization**: `MetaTKGR` initializes entity-specific parameters from global parameters via step (1), and fine-tunes them on support set via step (2) to optimize entity-specific parameters; In the **outer optimization**, `MetaTKGR` optimizes global parameters on query set via step (3) to learn good and robust global parameters.

generalized knowledge (i.e., learning to learn ability) by this bi-level optimization, as this procedure mimics the normal machine learning and inference process. For simplicity, to represent new entity $\tilde{e}$ along time, we denote $f_{\phi_{\tilde{e}}}$ as the temporal encoder parameterized by $\phi_{\tilde{e}}$, $\mathcal{L}(f_{\phi_{\tilde{e}}}, \mathcal{S}_{\tilde{e}})$ as model predictive loss on $\mathcal{S}_{\tilde{e}}$. Model-agnostic meta-learning (MAML) [11] can be utilized to fulfill this goal as follows:

$$\phi^* \longleftarrow \arg\min_{\phi} \mathbb{E}_{\tilde{e} \sim p(\tilde{\mathcal{E}})}\left[\mathcal{L}(f_{\phi_{\tilde{e}}}, \mathcal{Q}_{\tilde{e}})\right], \quad \text{where } \phi_{\tilde{e}} = \phi - \eta\frac{\partial\mathcal{L}(f_\phi, \mathcal{S}_{\tilde{e}})}{\partial\phi}, \tag{2}$$

where $\eta$ denotes inner-loop learning rate. However, this training strategy cannot guarantee good generalization ability over time (*challenge 2*), as they assume identical distributions between $\mathcal{S}_{\tilde{e}}$ and $\mathcal{Q}_{\tilde{e}}$, contradicting the facts that new entities are highly evolving over time on TKGs. To coherently resolve the two challenges, we advance the existing few-shot KG reasoning works by proposing:

- *Temporal encoder*, which learns the time-aware representation of each new entity $\tilde{e}$ by sampling and aggregating information from TKG neighbors on continuous domain.

- *Meta temporal reasoning*, which learns an optimal sampling and aggregating parameters via bi-level optimization (**inner optimization** and **outer optimization**). The learned parameters can be easily adapted to new entities and maintain temporal robustness.

### 3.3 Temporal Encoder

On temporal knowledge graphs, entities are evolving over time. The temporal encoder $f_\phi$ is to embed each entity $\tilde{e}$ into low-dimensional latent space at each time: $\mathbf{h}_{\tilde{e}}(t) \in \mathbb{R}^d$, where it can model the temporal pattern of $\tilde{e}$ along time. By doing so, it resolves the scarcity issues caused by few-shot links to some extent. Towards this goal, $f_\phi$ first samples temporal neighbors from TKGs, then attentively aggregates information from the temporal neighbors of each entity, which takes neighbor feature, relation feature and time feature into account.

---

**Algorithm 1:** Temporal Neighbor Sampler.

**Input:** New entity $\tilde{e}$, temporal knowledge graph $\mathcal{G}^t$, current timestamp $t$, neighbor budget $b$, time bound $\Delta t$.
**Output:** Temporal Neighbor $\mathcal{N}_{\tilde{e}}(t)$.
Initialize $Queue \leftarrow \tilde{e}$, $\mathcal{N}_{\tilde{e}}(t) \leftarrow \{\}$;
**while** $|\mathcal{N}_{\tilde{e}}(t)| < b$ *and Queue is not empty* **do**
    $e \leftarrow Queue.dequeue()$;
    **for** *each* $(e_s, r, e_o, t)$ *in* $\mathcal{G}^t$ **do**
        **if** $e_s == e$ *and* $t > t_{max} - \Delta t$ **then**
            Add $(e_s, r, t)$ to $\mathcal{N}_{\tilde{e}}(t)$, Add $e_s$ to $Queue$;
        **if** $e_o == e$ *and* $t > t_{max} - \Delta t$ **then**
            Add $(e_o, r, t)$ to $\mathcal{N}_{\tilde{e}}(t)$, Add $e_o$ to $Queue$;

---

To better represent few-shot entities, a wider range of neighbors should be considered, as the close neighbors around the new entities are usually insufficient. Conventionally, stacking several graph neural networks (GNNs)-based layers can integrate information from multi-hop neighbors [41, 16]. In few-shot cases, such layer-by-layer procedure faces difficulties. For one thing, the limited one-hop neighbors can dominate the aggregation process, as the information of multi-hop neighbors is all propagated from them to the target entity. Also, the size of neighbors grows exponentially with the increase of GNN layers, as they collect all neighbors in each hop without strategic selection,

causing severe efficiency issues. Instead, our temporal encoder first samples multi-hop neighbors via a time-bounded breadth-first-search algorithm, then aggregates information directly from the sampled neighbors in an attentive manner. As summarized in Algorithm 1, at each time, it selectively samples up to $b$ multi-hop temporal neighbors $\mathcal{N}_{\tilde{e}}(t)$ which have interactions within a recent time range $\Delta t$. Given the temporal neighbor $\mathcal{N}_{\tilde{e}}(t)$ for each new entity, temporal encoder $f_\phi$ represents a new entity as $\mathbf{h}_{\tilde{e}}(t)$ at time $t$:

$$\mathbf{h}_{\tilde{e}}^l(t) = \sigma \left( \sum_{(e_i, r_i, t_i) \in \mathcal{N}_{\tilde{e}}(t)} \alpha_{\tilde{e}, e_i} \left( \mathbf{h}_{e_i}^{l-1}(t_i) \mathbf{W} \right) \right), \tag{3}$$

where $l$ denotes the layer number, $\sigma(\cdot)$ denotes the activation function *ReLU*, $\alpha_{\tilde{e}, e_i}$ denotes the attention weight of entity $i$ to new entity $\tilde{e}$, and $\mathbf{W}$ is the trainable transformation matrix. To aggregate from history, $\alpha_{\tilde{e}, e_i}$ is supposed to be aware of entity feature, time delay and topology feature induced by relations. Thus, we design $\alpha_{\tilde{e}, i}$ as follows:

$$\alpha_{\tilde{e}, e_i} = \frac{\exp(q_{\tilde{e}, e_i})}{\sum_{(e_k, r_k, t_k) \in \mathcal{N}_{\tilde{e}}(t)} \exp(q_{\tilde{e}, e_k})}, \quad q_{\tilde{e}, e_i} = \mathbf{a} \left( \mathbf{h}_{\tilde{e}}^{l-1} \| \mathbf{h}_{e_i}^{l-1} \| \mathbf{h}_{r_i} \| \Phi(t - t_i) \right), \tag{4}$$

where $q_{\tilde{e}, e_i}$ measures the pairwise importance by considering the entity embedding, relation embedding and time embedding, $\mathbf{a} \in \mathbb{R}^{4d}$ is the shared parameter in the attention mechanism. Following [9] we adopt random Fourier features as time encoding $\Phi(\Delta t)$ to reflect the time difference.

### 3.4 Meta Temporal Reasoning

Let $\mathbf{h}_{\tilde{e}}^L(t)$ denote the representations of new entity $\tilde{e}$ produced by $f_\phi$ with $L$ layers. For each quadruple $(\tilde{e}, r, e, t)$, we employ a translation-based score function [2] to measure the correctness: $s(\tilde{e}, r, e, t; \phi) = -\|\mathbf{h}_{\tilde{e}}^L(t) + \mathbf{h}_r - \mathbf{h}_e^L(t)\|^2$. We first fine-tune $f_\phi$ on support set $\mathcal{S}_{\tilde{e}}$ for entity-specific parameters $f_{\phi_{\tilde{e}}}$, and then optimize training loss on query set $\mathcal{Q}_{\tilde{e}}$ to learn the global parameters shared by all entities. To calculate the predictive loss, since each set only contains positive quadruples, we perform negative sampling [2] to update MetaTKGR by training it to rank positive quadruples higher than negative ones. Specifically, taking the support set $\mathcal{S}_{\tilde{e}}$ of entity $\tilde{e}$ as an example, we construct a negative set $\mathcal{S}_{\tilde{e}}^- = \{(\tilde{e}, r, e^-, t) \text{ or } (e^-, r, \tilde{e}, t)\}$ by replace the ground truth entity $e$ with a corrupted entity $e^-$. We then use empirical hinge loss on the given set as follows:

$$\hat{\mathcal{L}}(f_{\phi_{\tilde{e}}}, \mathcal{S}_{\tilde{e}}) = \sum_{(e_s, r, e_o, t) \in \mathcal{S}_{\tilde{e}}} \sum_{(e_s, r, e_o, t)^- \in \mathcal{S}_{\tilde{e}}^-} \max \left( \gamma - s(e_s, r, e_o, t) + s(e_s, r, e_o, t)^-, 0 \right), \tag{5}$$

where $\gamma > 0$ is a margin value to distinguish positive and negative quadruples.

**Inner optimization**: For each task corresponding to new entity $\tilde{e} \sim p(\tilde{\mathcal{E}})$, we first adapt the global parameter $\phi$ for each new entity $\tilde{e}$ by minimizing the predictive loss on the support set $\mathcal{S}_{\tilde{e}}$:

$$\phi_{\tilde{e}} = \phi - \eta \frac{\partial \hat{\mathcal{L}}(f_\phi, \mathcal{S}_{\tilde{e}})}{\partial \phi}, \tag{6}$$

where $\eta$ is the inner loop learning rate. By Eq. 6, we simulate the adaption for new entities. Updating from an optimal global parameter $\phi$, the model is fine-tuned into entity-specific for new entities. Thus, it is crucial to design a meta-learning strategy to learn an optimal global parameter $\phi$ with a robust generalization ability over time.

**Outer optimization**: Since new entities evolve over time, it is not only $\mathcal{Q}_{\tilde{e}}$ but also each part of $\mathcal{Q}_{\tilde{e}}$ that follows

---

**Algorithm 2:** MetaTKGR: Meta-training.

**Input:** Temporal knowledge graph $\mathcal{G}^T \subseteq \mathcal{E}^T \times \mathcal{R} \times \mathcal{E}^T \times \mathcal{T}$, randomly initialized $\phi$.
**Output:** Learned global parameter $\phi$.
Simulate new entity set $\tilde{\mathcal{E}}$ from $\mathcal{E}^T$;
Train model parameter $\phi$ from many-shot entities $\mathcal{E}^T \setminus \tilde{\mathcal{E}}$;
Construct $\mathcal{S}_{\tilde{e}}$ and $\mathcal{Q}_{\tilde{e}}$ for each new entity $\tilde{e}$, where
  $\quad \mathcal{Q}_{\tilde{e}} = \{\mathcal{Q}_{\tilde{e}}^{(1)}, \mathcal{Q}_{\tilde{e}}^{(2)}, \cdots, \mathcal{Q}_{\tilde{e}}^{(M)}\}$;
**while** $\phi$ *not converge* **do**
  **for** *each time interval* $m$ **do**
    # **Outer optimization:**
    **for** *each new entity* $\tilde{e}$ **do**
      # **Inner optimization:**
      Calculate $\hat{\mathcal{L}}(f_\phi, \mathcal{S}_{\tilde{e}})$ on $\mathcal{S}_{\tilde{e}}$ by Eq. 5;
      Perform adaptation by Eq. 6 to update $\phi_{\tilde{e}}$;
      Calculate $\hat{\mathcal{L}}(f_{\phi_{\tilde{e}}}, \mathcal{Q}_{\tilde{e}}^{(m)})$ by Eq. 5;
    Calculate temporal adaptation regularizer by Eq. 8;
    Update $\phi^{(m)}$ by Eq. 9;
    $\phi \leftarrow \phi^{(m)}$

---

different distributions. Thus, instead of optimizing the meta-learner on the whole query set $\mathcal{Q}_{\tilde{e}}$ at

once by Eq. 2, we first adapt entity-specific parameters $\phi_{\tilde{e}}$ to recent facts, then gradually increase the difficulties by autoregressively adapting it to farther away future facts. Through this process, we are able to guide and stablize the training of global parameters via a temporal signal. Specifically, we split the time span of query sets into $M$ time intervals, where each new entity has a sequence of query set corresponding to different time intervals $\mathcal{Q}_{\tilde{e}} = \{\mathcal{Q}_{\tilde{e}}^{(1)}, \mathcal{Q}_{\tilde{e}}^{(2)}, \cdots, \mathcal{Q}_{\tilde{e}}^{(M)}\}$. We first adapt and optimize $\phi_{\tilde{e}}$ for each new entity on $\mathcal{Q}_{\tilde{e}}^{(1)}$ for $\phi^{(1)}$ encoding global knowledge for predictions in the 1-st time interval. Then we gradually adapt each $\phi_{\tilde{e}}$ fine-tuned from last time interval on farther away intervals until $\phi^{(M)}$ is learned, which maintain temporal robustness from time interval 1 to $M$.

We then discuss how to measure the feedback (predictive loss) from each adaptation. To simulate real scenarios, we assume query sets to follow different and unseen distributions. Taking adaptation from $m$-th time interval to $m + 1$-th time interval as example, we view $p(\phi^{(m)})$ as prior parameter distribution and aim to learn the posterior parameter distribution $q(\phi^{(m+1)})$ conditioning on query set in $m + 1$-th interval. The unseen query set distribution in $m + 1$-th interval prevent us to estimate $q(\phi^{(m+1)})$ by either using unbiased empirical loss or Bayes rules. To resolve this, we instead adopt the PAC-Bayes method [35], which allows one to learn a parameter posterior that fits the observations without knowing the observations distribution. We propose the following theorem to relate real predictive loss in new time interval with its empirical verson:

**Theorem 3.1** (**PAC-Bayes Generalization Bound on Temporal Adaptation**). *Let $\mathcal{D}^{(m+1)} = \bigcup_{\tilde{e} \in \tilde{\mathcal{E}}} \mathcal{Q}_{\tilde{e}}^{(m+1)}$ denote all query sets in $m + 1$-th time interval of all new entities from $\tilde{\mathcal{E}}$. For any $\delta \in (0, 1)$ and learned parameter distribution $p(\phi^{(m)})$ in last time interval, with probability at least $1 - \delta$ on $\mathcal{D}^{(m+1)}$:*

$$\mathcal{L}(f_{\phi^{(m)}}, \mathcal{D}^{(m+1)}) \leq \hat{\mathcal{L}}(f_{\phi^{(m)}}, \mathcal{D}^{(m+1)}) + \sqrt{\frac{\mathbb{KL}(q(\phi^{(m+1)}) \| p(\phi^{(m)})) + \log \frac{|\mathcal{D}^{(m+1)}|}{\delta}}{2|\mathcal{D}^{(m+1)}| - 1}}, \tag{7}$$

*where $\mathcal{L}(f_{\phi^{(m)}}, \mathcal{D}^{(m+1)}) = \mathbb{E}_{\tilde{e} \sim p(\tilde{\mathcal{E}})} \left[ \mathcal{L}(f_{\phi_{\tilde{e}}^{(m)}}, \mathcal{Q}_{\tilde{e}}^{(m+1)}) \right]$ denotes real predictive loss on new time interval with new and unknown data distribution, $\hat{\mathcal{L}}(f_{\phi^{(m)}}, \mathcal{D}^{(m+1)})$ denote the empirical version of the loss, and $\mathbb{KL}(\cdot \| \cdot)$ is the Kullback-Leibler divergence.*

Readers can refer to Appendix A.1 for proof. Thus, using Theorem 3.1, we can adapt $\phi^{(m)}$ to $m + 1$ time interval and estimate the feedback (predictive loss) for our meta-learning framework via the following *temporal adaptation regularizer*:

$$\mathcal{R}(f_{\phi^{(m)}}; \mathcal{D}^{(m+1)}) \triangleq \hat{\mathcal{L}}(f_{\phi^{(m)}}, \mathcal{D}^{(m+1)}) + \sqrt{\frac{\mathbb{KL}(q(\phi^{(m+1)}) \| p(\phi^{(m)})) + \log \frac{|\mathcal{D}^{(m+1)}|}{\delta}}{2|\mathcal{D}^{(m+1)}| - 1}}. \tag{8}$$

**Remark**. To interpret the temporal adaptation regularizer $\mathcal{R}(f_{\phi^{(m)}}; \mathcal{D}^{(m+1)})$, it can be viewed as a combination of empirical risk on the query set in the new time interval as well as a regularizer of parameter distribution across time intervals in form of KL-divergence. The regularizer along time domain can guarantee that global knowledge $\phi$ is trained by predictive losses from all time intervals without overfitting in specific time interval. Thus, we can improve the generalization ability of our meta-learner over time by the following update step by step, from $m = 1$ to $m = M$:

$$\phi^{(m+1)} = \phi^{(m)} - \beta \frac{\partial \mathcal{R}(f_{\phi^{(m)}}; \mathcal{D}^{(m+1)})}{\partial \phi}, \tag{9}$$

where $\phi^{(0)}$ can be obtained by training on existing entities. The meta-training phase of `MetaTKGR` is summarized in Algorithm 2. During meta-test phase, given a new entity $\tilde{e}$, we can first fine-tune $\phi$ on few-shot links, and then utilize $f_{\phi_{\tilde{e}}}$ to represent $\tilde{e}$ for future predictions.

# 4 Experiments

**Datasets.** We evaluate the proposed `MetaTKGR` framework on three public TKGs, where **YAGO** [34] and **WIKI** [22] stores time-varying facts and **ICEWS18** [3] is event-centric. They are collected from different time ranges and have different time units, which can validate our framework in the context of various types of evolution. *Notably, most new entities appear with few-shot facts,* validating the practical value of our proposed setting. Table 1 shows the detailed statistics. We report detailed dataset descriptions as well as entity links distribution in the Appendix A.2.

Table 2: The results of 3-shot temporal knowledge graph reasoning. Average results on 5 independent runs are reported. ∗ indicates the statistically significant improvements over the best baseline, with $p$-value smaller than 0.001. We report standard deviation in Appendix A.5

| Models | YAGO | | | | WIKI | | | | ICEWS18 | | | |
|---|---|---|---|---|---|---|---|---|---|---|---|---|
| | MRR | H@1 | H@3 | H@10 | MRR | H@1 | H@3 | H@10 | MRR | H@1 | H@3 | H@10 |
| TransE | 0.223 | 0.158 | 0.242 | 0.360 | 0.161 | 0.111 | 0.171 | 0.236 | 0.058 | 0.052 | 0.062 | 0.098 |
| TransR | 0.234 | 0.165 | 0.259 | 0.382 | 0.183 | 0.138 | 0.188 | 0.245 | 0.061 | 0.062 | 0.073 | 0.109 |
| RotatE | 0.241 | 0.182 | 0.278 | 0.409 | 0.232 | 0.171 | 0.223 | 0.284 | 0.078 | 0.074 | 0.082 | 0.128 |
| RE-NET | 0.261 | 0.210 | 0.298 | 0.410 | 0.261 | 0.210 | 0.251 | 0.331 | 0.232 | 0.139 | 0.241 | 0.369 |
| RE-GCN | 0.283 | 0.226 | 0.307 | 0.421 | 0.277 | 0.223 | 0.262 | 0.344 | 0.233 | 0.142 | 0.238 | 0.350 |
| LAN | 0.230 | 0.154 | 0.247 | 0.352 | 0.185 | 0.133 | 0.201 | 0.287 | 0.207 | 0.119 | 0.234 | 0.321 |
| I-GEN | 0.303 | 0.238 | 0.323 | 0.420 | 0.221 | 0.179 | 0.229 | 0.264 | 0.212 | 0.120 | 0.251 | 0.346 |
| T-GEN | 0.292 | 0.218 | 0.310 | 0.394 | 0.234 | 0.185 | 0.222 | 0.271 | 0.169 | 0.122 | 0.184 | 0.265 |
| MetaDyGNN | 0.350 | 0.270 | 0.379 | 0.511 | 0.309 | 0.238 | 0.309 | 0.459 | 0.307 | 0.216 | 0.309 | 0.469 |
| MetaTKGR | **0.370*** | **0.303*** | **0.416*** | **0.558*** | **0.329*** | **0.253*** | **0.335*** | **0.489*** | **0.335*** | **0.249*** | **0.340*** | **0.527*** |
| Gains (%) | *5.68* | *12.21* | *9.80* | *9.19* | *6.26* | *6.38* | *8.47* | *6.35* | *9.11* | *14.85* | *9.89* | *12.4* |

Table 3: The results of 1-shot and 2-shot experiment. We report complete results in Appendix A.5.

| Models | YAGO | | | | WIKI | | | | ICEWS18 | | | |
|---|---|---|---|---|---|---|---|---|---|---|---|---|
| | 1-shot | | 2-shot | | 1-shot | | 2-shot | | 1-shot | | 2-shot | |
| | MRR | H@10 | MRR | H@10 | MRR | H@10 | MRR | H@10 | MRR | H@10 | MRR | H@10 |
| TransE | 0.183 | 0.268 | 0.193 | 0.304 | 0.144 | 0.186 | 0.146 | 0.213 | 0.049 | 0.077 | 0.058 | 0.086 |
| TransR | 0.189 | 0.270 | 0.198 | 0.312 | 0.160 | 0.183 | 0.160 | 0.225 | 0.050 | 0.080 | 0.060 | 0.090 |
| RotatE | 0.215 | 0.280 | 0.210 | 0.359 | 0.175 | 0.190 | 0.201 | 0.268 | 0.068 | 0.098 | 0.070 | 0.091 |
| RE-NET | 0.221 | 0.304 | 0.233 | 0.390 | 0.212 | 0.259 | 0.239 | 0.294 | 0.185 | 0.250 | 0.200 | 0.341 |
| RE-GCN | 0.233 | 0.320 | 0.241 | 0.407 | 0.223 | 0.250 | 0.247 | 0.310 | 0.193 | 0.247 | 0.205 | 0.347 |
| LAN | 0.196 | 0.269 | 0.200 | 0.310 | 0.174 | 0.275 | 0.162 | 0.273 | 0.170 | 0.301 | 0.188 | 0.317 |
| I-GEN | 0.238 | 0.321 | 0.237 | 0.402 | 0.181 | 0.241 | 0.223 | 0.287 | 0.199 | 0.320 | 0.177 | 0.337 |
| T-GEN | 0.247 | 0.331 | 0.260 | 0.379 | 0.202 | 0.245 | 0.240 | 0.319 | 0.131 | 0.262 | 0.161 | 0.259 |
| MetaDyGNN | 0.269 | 0.396 | 0.316 | 0.496 | 0.241 | 0.371 | 0.271 | 0.390 | 0.249 | 0.420 | 0.269 | 0.441 |
| MetaTKGR | **0.294*** | **0.428*** | **0.356*** | **0.526*** | **0.277*** | **0.419*** | **0.309*** | **0.441*** | **0.295*** | **0.496*** | **0.301*** | **0.500*** |
| Gains (%) | *9.43* | *8.04* | *12.69* | *6.14* | *14.64* | *12.93* | *14.04* | *13.20* | *18.45* | *17.87* | *11.47* | *13.39* |

Table 1: Dataset statistics.

| Datasets | # Entities | # Relations | # Quadruples | Time Unit |
|---|---|---|---|---|
| YAGO | 10,623 | 10 | 201,090 | 1 year |
| WIKI | 12,554 | 24 | 669,935 | 1 year |
| ICEWS18 | 23,033 | 256 | 281,205 | 1 day |

**Baselines.** We compare nine state-of-the-art baselines from four related areas: 1) **TransE** [2], 2) **TransR** [33], 3) **RotatE** [46]: Translation distance based embedding methods for static knowledge graphs; 4) **RE-NET** [20], 5) **RE-GCN** [31]: Temporal knowledge graph embedding methods; 6) **LAN** [52]; 7) **I-GEN** [1]; 8) **T-GEN** [1]: Few-shot methods on static knowledge graph; 9)**MetaDyGNN** [65]: A meta-learning framework for few-shot link prediction on homogeneous graphs. We describe the baselines in detail in Appendix A.3.

**Experimental Setup.** Given the temporal knowledge graph, we first split the time duration into four with a ratio of 0.4:0.25:0.1:0.25 chronologically, then we collect the entities that firstly appear in each period as background/meta-training/meta-validation/meta-test entity set. We train our model on background set as initialization and simulate few-shot tasks on meta-training set. To construct few-shot tasks, we assume the first $K = 1, 2, 3$ quadruples of new entities are known and report the performance of the remaining quadruples in the future. For fair comparison, we keep the dimension of all embeddings as 128, and utilize pre-trained 1-shot TransE embeddings for initialization for models if applicable. We report detailed experimental setup, especially of MetaTKGR, in the Appendix A.4.

**Evaluation Protocol and Metrics.** For each prediction $(\tilde{e}, r, ?, t)$ or $(?, r, \tilde{e}, t)$, we use ranking scheme to evaluate the performance. Specifically, we rank all entities at the missing position in quadruples, and adopt mean reciprocal rank (*MRR*) and Hits at {1,3,10} (*H@{1,3,10}*) as evaluation metrics. It is worth noting that we measure the ranks in a filtered setting, where we filter other true quadruples existing in datasets, following [33, 46].

**Main Results.** Table 2 and Table 3 report overall result for $K = 3$ and $K = 1, 2$ few-shot experiments respectively. On average, MetaTKGR achieves 11.4% relative improvement over best baseline model, demonstrating the superiority of MetaTKGR in modeling new entity evolution in low-data regime. Baselines (*TransE, TransR, RotatE, RE-NET*) that do not optimize for new entities perform poorly on all datasets, although they are trained on both existing entities and new entities with few-shot links. *LAN, I-GEN, T-GEN* also produce unsatisfying results, despite the fact that they propose special designs to represent new entities. Note that they have worse performance in some cases than RE-NET that does not optimize for new entities, as they ignore the temporal information and perform poorly in future predictions. Compared with MetaDyGNN, MetaTKGR shows consistently better performance,

because our temporal encoder can handle multi-relational graphs in low-data regimes better, and our temporal meta-learning framework can produce more robust predictions.

**Performance of Prediction over Time**. Next, we study the performance of `MetaTKGR` over time. Figure 2 shows the performance comparisons of 3-shot predictions over different timestamps on the *YAGO* and *ICEWS18* datasets, measured by

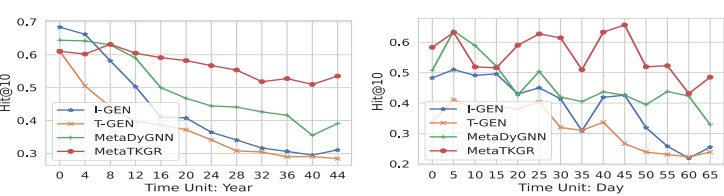

Figure 2: Performance of YAGO (left) and ICEWS18 (right) over time.

filtered $H@10$ metrics. `MetaTKGR` consistently beats all strong baselines. Although the compared baselines can optimize newly emerging entities, they perform poorly in wide time intervals because they largely ignore the temporal information and the distribution discrepancy caused by evolution. We notice that the relative gains of our model get more significant with increasing time steps. It illustrates that our temporal meta-learning framework can improve the generalization ability over time. Performance on ICEWS18 fluctuates more severely. This is expected since the evolution of ICEWS18 is not stable due to the much shorter time unit (day).

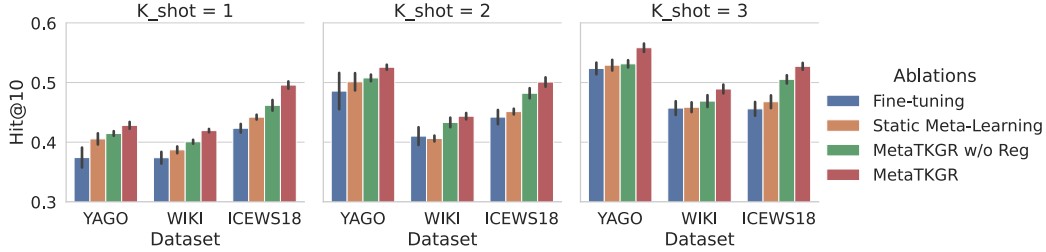

Figure 3: Ablation Studies, evaluated by filtered $Hit@10$.

**Ablation Study**. We evaluate performance improvements brought by the temporal meta-learning framework by following ablations: 1) *Fine-Tuning* is trained on existing entities without using any meta-learning strategy, and then fine-tuned on new entities; 2) *Static Meta-Learning* utilizes conventional MAML to train models; 3) `MetaTKGR w/o Regularizer` meta-trains model parameters on each time interval step by step, without explicitly optimizing the temporal adaptation regularizer. Figure 3 reports the results measured by filtered $H@10$. We can conclude that modeling the distribution discrepancy caused by the temporal evolution and optimizing the temporal adaptation regularizer can improve the performance in the future prediction. Also, simply fine-tuning the model parameters on new entities leads to poor results, as during background phase, the model does not extract generalized knowledge that can be easily adapted to new entities.

| Test | 1-Shot (Training) | | | 3-Shot (Training) | | |
|------|------|------|------|------|------|------|
|      | MRR  | H@1  | H@10 | MRR  | H@1  | H@10 |
| 1-S  | 0.294 | 0.240 | 0.428 | 0.281 | 0.242 | 0.403 |
| 3-S  | 0.354 | 0.297 | 0.540 | 0.370 | 0.303 | 0.558 |
| 5-S  | 0.377 | 0.360 | 0.569 | 0.389 | 0.369 | 0.591 |
| R-S  | 0.358 | 0.304 | 0.542 | 0.377 | 0.316 | 0.570 |

Table 4: Cross-shot results on YAGO.

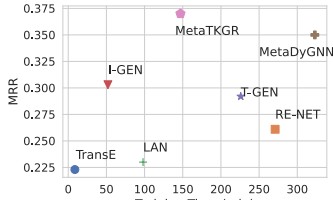

Figure 4: MRR over training time.

**Cross-shot Learning**. In reality, new entities are usually associated with various numbers of facts initially. Thus, the robustness of models on different numbers of shots during testing phase is critical. To simulate such scenario, we evaluate `MetaTKGR` by varying the number of shots including 1-, 3-, 5-, and random-shot (R-S: between 1 and 5) during meta-training and meta-test phases. Table 4 reports the cross-shot learning results. As expected, with the increase of observable shots during testing phase, the performance becomes better. The differences in the number of shots used for training do not significantly affect the results, demonstrating that `MetaTKGR` trained with a fixed number of shots performs robustly under the various number of shots during testing.

**Efficiency Analysis**. We train `MetaTKGR` and baseline models from scratch on both existing entities and new entities and compare the training time, for the 3-shot experiment on YAGO. Figure 4 shows that `MetaTKGR` significantly outperforms baseline models with reasonable training time. Compared with slow temporal models *RE-NET, MetaDyGNN* for knowledge graph reasoning, `MetaTKGR` is more efficient because 1) our temporal encoder can learn temporal entity embeddings via sampled temporal neighbors at each continuous timestamp without using RNNs; 2) our temporal neighborhood sampler can prevent the size of multi-hop neighbors from increasing exponentially.

## 5 Related Work

**Few-shot Learning on Knowledge Graph**. To alleviate data scarcity, various advanced techniques have emerged, such as transfer learning [37, 25], semi-supervised learning [5], domain adaptation [30, 27, 28, 26] and few-shot learning (FSL) [56, 67]. Few-shot learning aims at learning generalized knowledge from existing tasks to extract transferable priors for new tasks with few labeled samples, including metric learning based approaches [50, 45] and meta-learning based approaches [11, 39, 12, 29]. For knowledge graph reasoning, the success of few-shot learning has facilitated learning the representations of scarce entities [15, 52, 43, 1] and scarce relations [68, 58, 7, 36] respectively. In this paper, we are primarily interested in works to represent entities with few-shot facts, as relation set is relatively stable along time [44]. Given a new entity linked with few-shot facts, the neighborhood aggregator used in graph neural networks (GNNs) reduces the inductive bias via structural knowledge [21, 16, 41]. [15] computes the representations of few-shot entities by GNN-based neighboring aggregation scheme. [52, 43, 36] further extend it by utilizing attention mechanisms. However, those models are usually trained on both many-shot and few-shot entities, ending up being suboptimal for few-shot entities. To address it, a recent work [1] explores how to mate-train the neighborhood aggregator so as to effectively adapt global knowledge for few-shot knowledge reasoning. Deep graph learning has been attracting enormous interest recently [21, 16, 51, 61, 63, 64, 62, 60, 23]. More broadly on graphs, Meta-Graph [4], G-Meta [18] and MetaDyGNN [65] are proposed to utilize meta-learning for link prediction across multiple graphs, on static/temporal homogeneous graphs. However, most works are designed for few-shot learning on static (knowledge) graphs. How to aggregate neighbors considering time factors and optimize for long-term performance are unsolved. MetaDyGNN [65] is a recent framework combining MAML with TGAT [9] on temporal homogeneous graph. But it did not consider distribution difference between few-shot facts and future facts caused by the evolution of new entities either, leading to increasingly poor performance over time, especially on temporal knowledge graphs with more complicated structures.

**Temporal Knowledge Graph Reasoning**. Temporal knowledge graphs (TKGs) store time-varying facts in the real-world. Temporal knowledge graph reasoning aims to predict missing facts at a certain time in the future. It is mostly formulated as measuring the correctness of factual samples and negative samples by specially designed score functions [2, 33, 49, 46, 13]. Compared with static KG reasoning tasks [19], the main challenge lies in how to incorporate time information into the representation process. Several embedding-based methods have been proposed. They encode time-dependent information of entities and relations by decoupling embeddings into static component and time-varying component [59, 14], utilizing recurrent neural networks (RNNs) to adaptively learn the dynamic evolution from historical fact sequence [20, 57], or learning a sequence of evolving representations from discrete knowledge graph snapshots [17, 32, 20]. However, all of the existing temporal KG reasoning models aim to extrapolate future facts among existing entities, and how to predict future facts specifically for new emerging entities is largely under-explored.

## 6 Conclusion

We study a realistic but underexplored few-shot temporal knowledge graph reasoning problem, which aims at predicting future facts for newly emerging entities with a few facts. To this end, we propose a novel Meta Temporal Knowledge Graph Reasoning framework `MetaTKGR`. It meta-learns the global knowledge of sampling and aggregating temporal neighbors, which can be adapted quickly to new entities for future prediction. Such procedure is gradually guided by the performance on predicting future facts, from near time intervals to far away ones. We further theoretically analyze and propose a temporal adaptation regularizer to stabilize and generalize the learned knowledge on future tasks. We empirically validate the effectiveness of MetaTKGR on three real-world temporal knowledge graphs, on which the proposed framework significantly outperforms an extensive set of SOTA baselines.

## Acknowledgments and Disclosure of Funding

The authors would like to thank Tianshi Wang, Yuchen Yan, Jialu Wang for their helpful comments and discussion. The authors would also like to thank the anonymous reviewers of NeurIPS for valuable comments and suggestions. Research reported in this paper was sponsored in part by DARPA award HR001121C0165, DARPA award HR00112290105, Basic Research Office award HQ00342110002, and the Army Research Laboratory under Cooperative Agreement W911NF-17-20196.

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
