# Supplementary Material of Learning to Sample and Aggregate: Few-shot Reasoning over Temporal Knowledge Graphs

**Ruijie Wang[1], Zheng Li[2], Dachun Sun[1], Shengzhong Liu[1], Jinning Li[1],**

**Bing Yin[2], Tarek Abdelzaher[1]**
[1]University of Illinois at Urbana Champaign, IL, USA
[2]Amazon.com Inc, CA, USA
{ruijiew2, dsun18, sl29, jinning4, zaher}@illinois.edu
{amzzhe, alexbyin}@amazon.com

## A  Appendix

The supplementary material is structured as follows:

- Section A.1 gives the proof and analysis of Theorem 3.1;
- Section A.2 introduces the datasets and their statistics in detail;
- Section A.3 introduces the baselines utilized in experiments;
- Section A.4 discusses the experimental setup of baseline models as well as MetaTKGR;
- Section A.5 reports detailed experiment performance with statistical test results;

### A.1  Statements, Proof and Analysis of Theorem 3.1

**Theorem A.1 (PAC-Bayes Generalization Bound on Temporal Adaptation).** *Let* $\mathcal{D}^{(m+1)} = \bigcup_{\tilde{e} \in \tilde{\mathcal{E}}} \mathcal{Q}_{\tilde{e}}^{(m+1)}$ *denote all query set in* $m+1$*-th time interval of all new entities from* $\tilde{\mathcal{E}}$*. For any* $\delta \in (0,1)$ *and learned parameter distribution* $p(\phi^{(m)})$ *in last time interval, with probability at least* $1 - \delta$ *on* $\mathcal{D}^{(m+1)}$*:*

$$\mathcal{L}(f_{\phi^{(m)}}, \mathcal{D}^{(m+1)}) \leq \hat{\mathcal{L}}(f_{\phi^{(m)}}, \mathcal{D}^{(m+1)}) + \sqrt{\frac{\mathbb{KL}(q(\phi^{(m+1)}) \| p(\phi^{(m)})) + \log \frac{|\mathcal{D}^{(m+1)}|}{\delta}}{2|\mathcal{D}^{(m+1)}| - 1}}, \quad (1)$$

*where* $\mathcal{L}(f_{\phi^{(m)}}, \mathcal{D}^{(m+1)}) = \mathbb{E}_{\tilde{e} \sim p(\tilde{E})} \left[ \mathcal{L}(f_{\phi_{\tilde{e}}^{(m)}}, \mathcal{Q}_{\tilde{e}}^{(m+1)}) \right]$ *denotes real predictive loss on new time interval with new and unknown data distribution,* $\hat{\mathcal{L}}(f_{\phi^{(m)}}, \mathcal{D}^{(m+1)})$ *denote the empirical version of the loss, and* $\mathbb{KL}(\cdot \| \cdot)$ *is the Kullback-Leibler divergence.*

*Proof.* As a realistic scenario, $\mathcal{D}^{(m+1)}$ in the new time interval follow different distribution from those in the last time interval, i.e., $m$-th time interval, and such new distribution is unknown at advance. To improve the generalization ability over time, we gradually adapt model parameters learned from last time interval $\phi^{(m)}$ to next time interval, resulting in updated parameters $\phi^{(m+1)}$. Formally, we view $p(\phi^{(m)})$ as prior parameter distribution and aim to learn the posterior parameter distribution $q(\phi^{(m+1)})$ conditioning on $\mathcal{D}^{(m+1)}$. The unseen distribution of $\mathcal{D}^{(m+1)}$ prevents us from estimating $q(\phi^{(m+1)})$ by either using unbiased empirical loss or Bayes rules. Therefore, for

36th Conference on Neural Information Processing Systems (NeurIPS 2022).

convinience of discussion, we first define the difference of real predictive loss and its empirical extimation as follows:

$$\Delta\mathcal{L} = \mathcal{L}(f_{\phi^{(m)}}, \mathcal{D}^{(m+1)}) - \hat{\mathcal{L}}(f_{\phi^{(m)}}, \mathcal{D}^{(m+1)}). \tag{2}$$

We are interested at the relation of $\Delta\mathcal{L}$ and the distribution discrenpency between $p(\phi^{(m)})$ and $p(\phi^{(m+1)})$. Towards this goal, following [9, 10], we construct the following function:

$$f(\mathcal{D}^{(m+1)}) = 2(|\mathcal{D}^{(m+1)}| - 1)\mathbb{E}_{\phi\sim q(\phi^{(m+1)})}\left[(\Delta\mathcal{L})^2\right] - \mathbb{KL}(q(\phi^{(m+1)})\|p(\phi^{(m)})). \tag{3}$$

Next, using Markov's inequality, we have:

$$p(f(\mathcal{D}^{(m+1)}) > \epsilon) = p(e^{f(\mathcal{D}^{(m+1)})} > e^{\epsilon}) \leq \frac{\mathbb{E}_{\tilde{e}}\left[e^{f(\mathcal{D}^{(m+1)})}\right]}{e^{\epsilon}}, \tag{4}$$

where $\mathbb{E}_{\tilde{e}}\left[e^{f(\mathcal{D}^{(m+1)})}\right]$ denotes the expectation of $e^{f(\mathcal{D}^{(m+1)})}$ w.r.t. new entity distribution. To upper bound the expectation, we have the following inequality:

$$f(\mathcal{D}^{(m+1)}) = 2(|\mathcal{D}^{(m+1)}| - 1)\mathbb{E}_{\phi\sim q(\phi^{(m+1)})}\left[(\Delta\mathcal{L})^2\right] - \mathbb{KL}(q(\phi^{(m+1)})\|p(\phi^{(m)})) \tag{5}$$

$$= \mathbb{E}_{\phi\sim q(\phi^{(m+1)})}\left[\log\left(e^{2(|\mathcal{D}^{(m+1)}|-1)(\Delta\mathcal{L})^2}\frac{q(\phi^{(m+1)})}{p(\phi^{(m)})}\right)\right] \tag{6}$$

$$\leq \log\left(\mathbb{E}_{\phi\sim q(\phi^{(m+1)})}\left[e^{2(|\mathcal{D}^{(m+1)}|-1)(\Delta\mathcal{L})^2}\frac{q(\phi^{(m+1)})}{p(\phi^{(m)})}\right]\right) \tag{7}$$

$$= \log\left(\mathbb{E}_{\phi\sim p(\phi^{(m)})}\left[e^{2(|\mathcal{D}^{(m+1)}|-1)(\Delta\mathcal{L})^2}\right]\right), \tag{8}$$

where Jensen's inequality is utilzied to derive the inequality. Therefore, we have

$$\mathbb{E}_{\tilde{e}}\left[e^{f(\mathcal{D}^{(m+1)})}\right] \leq \mathbb{E}_{\tilde{e}}\mathbb{E}_{\phi\sim p(\phi^{(m)})}\left[e^{2(|\mathcal{D}^{(m+1)}|-1)(\Delta\mathcal{L})^2}\right] \tag{9}$$

$$= \mathbb{E}_{\phi\sim p(\phi^{(m)})}\mathbb{E}_{\tilde{e}}\left[e^{2(|\mathcal{D}^{(m+1)}|-1)(\Delta\mathcal{L})^2}\right], \tag{10}$$

we switch the order of expectations because $p(\phi^{(m)})$ is independent to $\mathcal{D}^{(m+1)}$. Next, based on Hoeffding's inequality, we have:

$$p(\Delta\mathcal{L} > \epsilon) \leq e^{-2|\mathcal{D}^{(m+1)}|\epsilon^2}, \tag{11}$$

and we can further derive the following inequality:

$$\mathbb{E}_{\tilde{e}}\left[e^{f(\mathcal{D}^{(m+1)})}\right] \leq \mathbb{E}_{\phi\sim p(\phi^{(m)})}\mathbb{E}_{\tilde{e}}\left[e^{2(|\mathcal{D}^{(m+1)}|-1)(\Delta\mathcal{L})^2}\right] \leq |\mathcal{D}^{(m+1)}|. \tag{12}$$

Combining Eq. 12 and Eq. 4, we get:

$$p(f(\mathcal{D}^{(m+1)}) > \epsilon) \leq \frac{|\mathcal{D}^{(m+1)}|}{e^{\epsilon}} = \delta, \tag{13}$$

where $\delta = |\mathcal{D}^{(m+1)}|/e^{\epsilon}$. Therefore, with probability of at least $1 - \delta$, we have that for all $\phi^{(m+1)}$:

$$f(\mathcal{D}^{(m+1)}) = 2(|\mathcal{D}^{(m+1)}| - 1)\mathbb{E}_{\phi\sim q(\phi^{(m+1)})}\left[(\Delta\mathcal{L})^2\right] - \mathbb{KL}(q(\phi^{(m+1)})\|p(\phi^{(m)})) \tag{14}$$

$$\leq \frac{\log|\mathcal{D}^{(m+1)}|}{\delta}. \tag{15}$$

Further, by utilizing Jensen's inequality again, we have:

$$\left(\mathbb{E}_{\phi\sim q(\phi^{(m+1)})}\left[(\Delta\mathcal{L})\right]\right)^2 \leq \mathbb{E}_{\phi\sim q(\phi^{(m+1)})}\left[(\Delta\mathcal{L})^2\right] \tag{16}$$

$$\leq \frac{\mathbb{KL}(q(\phi^{(m+1)})\|p(\phi^{(m)})) + \frac{\log|\mathcal{D}^{(m+1)}|}{\delta}}{2(|\mathcal{D}^{(m+1)}| - 1}. \tag{17}$$

Subsituting definition of $\Delta\mathcal{L}$ in Eq. 17, we proof the PAC-Bayes Generalization Bound on Temporal Adaptation:

$$\mathcal{L}(f_{\phi^{(m)}}, \mathcal{D}^{(m+1)}) \leq \hat{\mathcal{L}}(f_{\phi^{(m)}}, \mathcal{D}^{(m+1)}) + \sqrt{\frac{\mathbb{KL}(q(\phi^{(m+1)})\|p(\phi^{(m)})) + \log\frac{|\mathcal{D}^{(m+1)}|}{\delta}}{2|\mathcal{D}^{(m+1)}| - 1}} \quad (18)$$

Based on Eq. 18, we define the following *temporal adaptation regularizer*:

$$\mathcal{R}(f_{\phi^{(m)}}; \mathcal{D}^{(m+1)}) \triangleq \hat{\mathcal{L}}(f_{\phi^{(m)}}, \mathcal{D}^{(m+1)}) + \sqrt{\frac{\mathbb{KL}(q(\phi^{(m+1)})\|p(\phi^{(m)})) + \log\frac{|\mathcal{D}^{(m+1)}|}{\delta}}{2|\mathcal{D}^{(m+1)}| - 1}}. \quad (19)$$

$\square$

**Remark**. To interpret the temporal adaptation regularizer $\mathcal{R}(f_{\phi^{(m)}}; \mathcal{D}^{(m+1)})$, it can be viewed as a combination of empirical risk on the query set in the new time interval as well as a regularizer of parameter distribution across time intervals in form of KL-divergence. The regularizer along time domain can guarantee that global knowledge $\phi$ is trained by predictive losses from all time intervals without overfitting in specific time interval. Such regularizre provides a stable feedback signal to guide the outer optimization phase. Thus, we can improve the generalization ability of our meta-learner over time by the following update step by step,

## A.2 Datasets

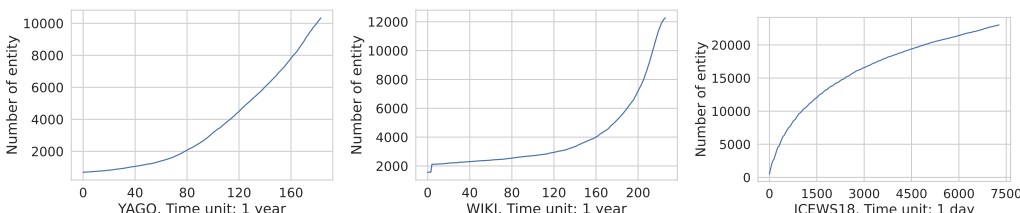

Figure 1: Number of entities over time. New entities continuously emerge on three public TKGs.

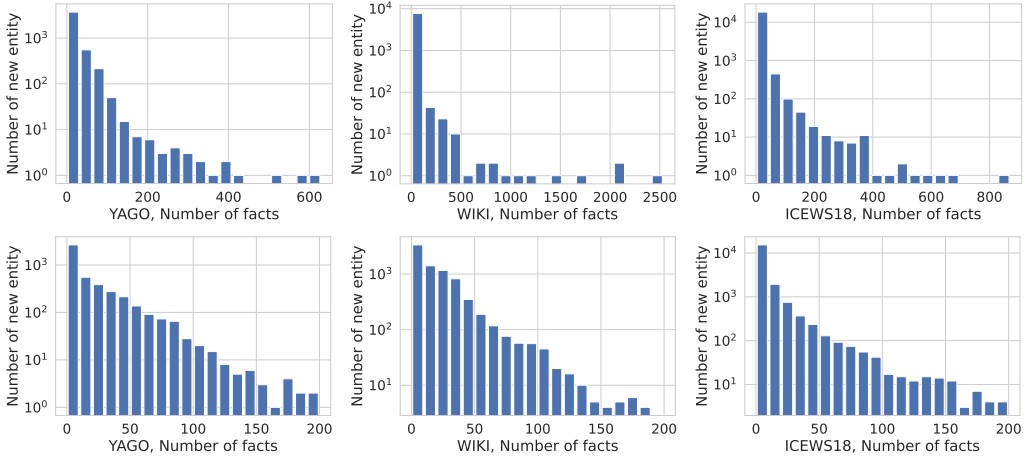

Figure 2: Distribution for new entity occurrences. The figures in first row show the complete distribution, and the figures in second row show the detailed distribution between [0,200]. Most of new entities are associated with limited amount of facts.

**Dataset Information**. We validate the proposed MetaTKGR framework on three public temporal knowledge graphs: 1) **YAGO** [8]: YAGO is a collection from the Wikipedias in multiple languages. 2)**WIKI** [5]: WIKI is a newly collected temporal knowledge graph of Wikipedias. 3)**ICEWS18** [3]: Integrated Crisis Early Warning System (ICEWS18) is the collection of coded interactions between

socio-political actors which are extracted from news articles. ICEWS18 is collected from 1/1/2018 to 10/31/2018, and the minimum time unit is one day; YAGO and WIKI span much longer periods (184 years and 232 years respectively), with one year as minimum time units. They can validate our temporal framework within different time ranges. Notably, ICEWS18 represents temporal facts as quadruples $(e_s, r, e_o, t)$, and WIKI, YAGO datasets consist of facts in format $(e_s, r, e_o, [t_s, t_e])$, where each fact is associated with a valid time range from start time $t_s$ to end time $t_e$. We follow [4] to preprocess WIKI and YAGO. Specifically, we preprocess the format such that each fact is converted to a sequence $\{(e_s, r, e_o, t_s), (e_s, r, e_o, t_s + 1), \cdots, (e_s, r, e_o, t_e)\}$ from $t_s$ to $t_e$, with the minimum time unit as one step. Noisy events of early years are removed (before 1786 for WIKI and 1830 for YAGO). Figure 1 shows the amount of new entities appearing over time. New entities continuously join the TKGs on three public TKGs, **e.g.,** $41.7$**,** $61.3\%$**,** $9.8\%$ **of entities on YAGO, WIKI, and ICEWS18 are new entities that firstly appear in the last** $25\%$ **time steps**. We further investigate the amount of facts associated with those new entities. Figure 2 shows the corresponding distributions. **Most new entities are associated with a limited amount of facts.** Figure 1 and Figure 2 validate the practical value of our proposed setting, which aims to predict the future facts of newly emerging entities based on initial few-shot observations.

**Splitting Scheme**. Given the temporal knowledge graph, we first split the time duration into four parts with a ratio of 0.4:0.25:0.1:0.25 chronologically, then we collect the entities that firstly appear in each period as well as the associated facts as training/meta-training/meta-validation/meta-test set. For the nodes in the meta-validation and meta-test set, we assume that their first $K = 1, 2, 3$ facts are known and can be used to adapt parameters for meta-learning based methods or train/compute new entity representations for non-meta ones. The remaining facts are utilized to evaluate performance of baseline models and MetaTKGR. Detailed setup is introduced in Appendix A.4.

## A.3 Baselines

We describe the baseline models utilized in the experiments in detail:

- **TransE** [2] is a translation-based embedding model, where both entities and relations are represented as vectors in the latent space. The relation is utilized as a translation operation between the subject and the object entity;
- **TransR** [7] advances TransE by optimizing modeling of n-n relations, where each entity embedding can be projected to hyperplanes defined by relations;
- **RotatE** [11] represents entities as complex vectors and relations as rotation operations in a complex vector space;
- **RE-NET** [4] is a generative model to predict future facts on temporal knowledge graphs, which employs a recurrent neural network to model the entity evolution, and utilizes a neighborhood aggregator to consider the connection of facts at the same time intervals;
- **RE-GCN** [6] learns the temporal representations of both entities and relations by modeling the KG sequence recurrently;
- **LAN** [12] computes the embedding of entities by GNN-based neighboring aggregation scheme, and attention mechanisms are utilized to consider relations with neighboring information for new entities;
- **I-GEN** [1] utilizes meta-learning technique to learn the representations of new entities, which is achieved by aggregating information from neighbors attentively;
- **T-GEN** [1] further extends I-GEN by optimizing predictions for unseen-unseen links among unseen users. A stochastic inference is proposed to model the randomness of such links;
- **MetaDyGNN** [13] is a recent work modeling the links predictions for new nodes on homogeneous graphs. A hierarchical meta-learner is proposed to better extract global knowledge which is beneficial for link prediction task.

## A.4 Experimental Setup

**Baseline Setup**. For static knowledge graph reasoning methods, i.e., TransE, TransR, and RotatE, we ignore all time information in quadruples, and view temporal knowledge graphs as static, cumulative ones. Then we train the models with training set, meta-training set as well as first $K$ triples of each

Table 1: The results of 1-shot temporal knowledge graph reasoning.

| Models | YAGO | | | | WIKI | | | | ICEWS18 | | | |
|---|---|---|---|---|---|---|---|---|---|---|---|---|
| | MRR | H@1 | H@3 | H@10 | MRR | H@1 | H@3 | H@10 | MRR | H@1 | H@3 | H@10 |
| TransE | 0.183 | 0.136 | 0.196 | 0.268 | 0.144 | 0.118 | 0.139 | 0.186 | 0.049 | 0.030 | 0.053 | 0.077 |
| TransE (std) | ± 0.008 | ± 0.005 | ± 0.004 | ±0.007 | ± 0.005 | ± 0.008 | ± 0.010 | ±0.011 | ± 0.004 | ± 0.004 | ± 0.003 | ±0.005 |
| TransR | 0.189 | 0.132 | 0.208 | 0.270 | 0.160 | 0.129 | 0.145 | 0.183 | 0.050 | 0.034 | 0.056 | 0.080 |
| TransR (std) | ± 0.007 | ± 0.011 | ± 0.007 | ±0.009 | ± 0.008 | ± 0.012 | ± 0.013 | ±0.009 | ± 0.005 | ± 0.006 | ± 0.004 | ±0.003 |
| RotatE | 0.215 | 0.143 | 0.217 | 0.280 | 0.175 | 0.139 | 0.153 | 0.190 | 0.068 | 0.053 | 0.062 | 0.098 |
| RotatE (std) | ± 0.005 | ± 0.009 | ± 0.012 | ±0.011 | ± 0.009 | ± 0.013 | ± 0.012 | ±0.008 | ± 0.006 | ± 0.004 | ± 0.010 | ±0.010 |
| RE-NET | 0.221 | 0.159 | 0.225 | 0.304 | 0.212 | 0.178 | 0.183 | 0.259 | 0.185 | 0.129 | 0.209 | 0.250 |
| RE-NET (std) | ± 0.006 | ± 0.008 | ± 0.004 | ±0.007 | ± 0.010 | ± 0.008 | ± 0.009 | ±0.010 | ± 0.013 | ± 0.011 | ± 0.012 | ±0.009 |
| LAN | 0.196 | 0.145 | 0.210 | 0.269 | 0.174 | 0.126 | 0.185 | 0.275 | 0.170 | 0.086 | 0.159 | 0.301 |
| LAN (std) | ± 0.015 | ± 0.015 | ± 0.019 | ±0.020 | ± 0.016 | ± 0.015 | ± 0.019 | ±0.014 | ± 0.021 | ± 0.019 | ± 0.022 | ±0.019 |
| I-GEN | 0.238 | 0.195 | 0.243 | 0.321 | 0.181 | 0.156 | 0.166 | 0.241 | 0.199 | 0.101 | 0.217 | 0.320 |
| I-GEN (std) | ± 0.007 | ± 0.005 | ± 0.009 | ±0.011 | ± 0.008 | ± 0.006 | ± 0.011 | ±0.013 | ± 0.021 | ± 0.019 | ± 0.022 | ±0.019 |
| T-GEN | 0.247 | 0.199 | 0.266 | 0.331 | 0.202 | 0.167 | 0.189 | 0.245 | 0.131 | 0.072 | 0.139 | 0.262 |
| T-GEN (std) | ± 0.023 | ± 0.019 | ± 0.022 | ±0.031 | ± 0.019 | ± 0.013 | ± 0.022 | ±0.024 | ± 0.031 | ± 0.026 | ± 0.029 | ±0.032 |
| MetaDyGNN | 0.269 | 0.219 | 0.297 | 0.396 | 0.241 | 0.176 | 0.271 | 0.371 | 0.249 | 0.179 | 0.269 | 0.420 |
| MetaDyGNN (std) | ± 0.010 | ± 0.008 | ± 0.011 | ±0.015 | ± 0.013 | ± 0.010 | ± 0.013 | ±0.016 | ± 0.012 | ± 0.009 | ± 0.014 | ±0.018 |
| MetaTKGR | **0.294*** | **0.240*** | **0.319*** | **0.428*** | **0.277*** | **0.203*** | **0.3071*** | **0.419*** | **0.295*** | **0.207*** | **0.309*** | **0.496*** |
| MetaTKGR (std) | ± 0.012 | ± 0.008 | ± 0.014 | ±0.009 | ± 0.015 | ± 0.011 | ± 0.013 | ±0.010 | ± 0.007 | ± 0.019 | ± 0.016 | ±0.012 |

Table 2: The results of 2-shot temporal knowledge graph reasoning.

| Models | YAGO | | | | WIKI | | | | ICEWS18 | | | |
|---|---|---|---|---|---|---|---|---|---|---|---|---|
| | MRR | H@1 | H@3 | H@10 | MRR | H@1 | H@3 | H@10 | MRR | H@1 | H@3 | H@10 |
| TransE | 0.193 | 0.139 | 0.204 | 0.304 | 0.146 | 0.122 | 0.146 | 0.213 | 0.058 | 0.042 | 0.054 | 0.086 |
| TransE (std) | ± 0.007 | ± 0.006 | ± 0.009 | ±0.013 | ± 0.006 | ± 0.005 | ± 0.007 | ±0.009 | ± 0.005 | ± 0.004 | ± 0.004 | ±0.005 |
| TransR | 0.198 | 0.140 | 0.217 | 0.312 | 0.160 | 0.131 | 0.170 | 0.225 | 0.060 | 0.054 | 0.061 | 0.090 |
| TransR (std) | ± 0.007 | ± 0.005 | ± 0.009 | ±0.011 | ± 0.005 | ± 0.004 | ± 0.008 | ±0.009 | ± 0.003 | ± 0.004 | ± 0.006 | ±0.007 |
| RotatE | 0.210 | 0.162 | 0.244 | 0.359 | 0.201 | 0.160 | 0.214 | 0.268 | 0.070 | 0.065 | 0.068 | 0.091 |
| RotatE (std) | ± 0.010 | ± 0.006 | ± 0.010 | ±0.009 | ± 0.009 | ± 0.006 | ± 0.005 | ±0.008 | ± 0.004 | ± 0.003 | ± 0.008 | ±0.008 |
| RE-NET | 0.233 | 0.220 | 0.281 | 0.390 | 0.239 | 0.191 | 0.238 | 0.294 | 0.200 | 0.109 | 0.214 | 0.341 |
| RE-NET (std) | ± 0.008 | ± 0.007 | ± 0.008 | ±0.010 | ± 0.005 | ± 0.007 | ± 0.008 | ±0.013 | ± 0.010 | ± 0.012 | ± 0.014 | ±0.019 |
| LAN | 0.200 | 0.144 | 0.209 | 0.310 | 0.162 | 0.108 | 0.178 | 0.273 | 0.188 | 0.089 | 0.176 | 0.317 |
| LAN (std) | ± 0.007 | ± 0.008 | ± 0.004 | ±0.006 | ± 0.010 | ± 0.006 | ± 0.008 | ±0.011 | ± 0.008 | ± 0.009 | ± 0.011 | ±0.013 |
| I-GEN | 0.237 | 0.216 | 0.291 | 0.402 | 0.223 | 0.185 | 0.230 | 0.287 | 0.177 | 0.092 | 0.212 | 0.337 |
| I-GEN (std) | ± 0.011 | ± 0.010 | ± 0.014 | ±0.017 | ± 0.009 | ± 0.009 | ± 0.012 | ±0.013 | ± 0.010 | ± 0.014 | ± 0.013 | ±0.020 |
| T-GEN | 0.260 | 0.200 | 0.278 | 0.379 | 0.240 | 0.193 | 0.248 | 0.319 | 0.161 | 0.111 | 0.185 | 0.259 |
| T-GEN (std) | ± 0.029 | ± 0.020 | ± 0.030 | ±0.032 | ± 0.023 | ± 0.019 | ± 0.022 | ±0.028 | ± 0.032 | ± 0.027 | ± 0.030 | ±0.031 |
| MetaDyGNN | 0.316 | 0.265 | 0.332 | 0.496 | 0.271 | 0.225 | 0.287 | 0.390 | 0.269 | 0.184 | 0.267 | 0.441 |
| MetaDyGNN (std) | ± 0.009 | ± 0.006 | ± 0.009 | ±0.011 | ± 0.010 | ± 0.009 | ± 0.014 | ±0.018 | ± 0.011 | ± 0.009 | ± 0.011 | ±0.015 |
| MetaTKGR | **0.356** | **0.284** | **0.372** | **0.526** | **0.309** | **0.249** | **0.325** | **0.441** | **0.300** | **0.208** | **0.311** | **0.500** |
| MetaTKGR (std) | ± 0.013 | ± 0.007 | ± 0.012 | ±0.010 | ± 0.014 | ± 0.013 | ± 0.012 | ±0.013 | ± 0.011 | ± 0.010 | ± 0.012 | ±0.014 |

new entity in meta-validation and meta-test sets. For temporal knowledge graph reasoning methods RE-NET and RE-GCN, we train the models with training set, meta-training set as well as first $K$ quadruples of each new entity in meta-validation and meta-test sets. For baselines that optimize few-shot entities on static knowledge graphs, we ignore all time information in quadruples. We train them via training set and meta-training set, and adapt parameters via the first $K$ quadruples of each new entity in meta-validation and meta-test sets. As for MetaDyGNN, since it is designed for homogeneous graphs, we adopt score function of TransE in the framework to support it on temporal knowledge graphs. Similarly, training and meta-training sets are utilized to train initial model parameters, then the few-shot facts of new entities are used to adapt entity-specific model parameters. For fair comparisons, we keep the dimension of all embeddings as 128, we feed pre-trained 1-shot TransE embeddings to those that require initial entity/relation embeddings, and we train all baseline models and MetaTKGR on same GPUs (GeForce RTX 3090) and CPUs (AMD Ryzen Threadripper 3970X 32-Core Processor).

**MetaTKGR Setup**. We use training set to train MetaTKGR for parameter initialization, then we simulate few-shot tasks by utilizing meta-training set to adapt model parameters and further improve the temporal generalization ability. For new entities, we use initial $K = 1, 2, 3$ facts to fine-tune entity-specific models. During evaluation, we tune hyperparameters based on *MRR* on meta-validation set, and report the performance on the remaining facts on meta-test set. Next, we report the choices of hyperparameters. For model training, we utilize Adam optimizer, and set maximum number of epochs as 50. We set batch size as 20, the dimension of all embeddings as 128, and dropout rate as 0.5. For the sake of efficiency, we set the neighbor budget $b$ of temporal neighbor sampler as 16, and employ 1 neighborhood aggregation layer in temporal encoder. We divide query sets into 3 time intervals to simulate the real scenario. We perform a single step of gradient descent for inner loop optimization. We mainly tune margin value $\gamma$ in score functions in range $\{0.3, 0.4, 0.5, 0.6, 0.7\}$, inner/outer loop learning rate $\eta$ and $\beta$ in range $\{0.01, 0.005, 0.001, 0.0005, 0.0001, 0.00005, 0.00001\}$. For YAGO and ICEWS18, we set $\gamma = 0.5$, $\eta = \beta = 0.0001$. For WIKI, we set $\gamma = 0.4$, $\eta = 0.00005$, and $\beta = 0.0001$.

Table 3: The results of 3-shot temporal knowledge graph reasoning.

| Models | YAGO | | | | WIKI | | | | ICEWS18 | | | |
|---|---|---|---|---|---|---|---|---|---|---|---|---|
| | MRR | H@1 | H@3 | H@10 | MRR | H@1 | H@3 | H@10 | MRR | H@1 | H@3 | H@10 |
| TransE | 0.223 | 0.158 | 0.242 | 0.360 | 0.161 | 0.111 | 0.171 | 0.236 | 0.058 | 0.052 | 0.062 | 0.098 |
| TransE (std) | ± 0.006 | ± 0.004 | ± 0.007 | ±0.011 | ± 0.007 | ± 0.008 | ± 0.010 | ±0.011 | ± 0.004 | ± 0.002 | ± 0.007 | ±0.006 |
| TransR | 0.234 | 0.165 | 0.259 | 0.382 | 0.183 | 0.138 | 0.188 | 0.245 | 0.061 | 0.062 | 0.073 | 0.109 |
| TransR (std) | ± 0.009 | ± 0.007 | ± 0.010 | ±0.013 | ± 0.004 | ± 0.003 | ± 0.006 | ±0.008 | ± 0.007 | ± 0.003 | ± 0.005 | ±0.008 |
| RotatE | 0.241 | 0.182 | 0.278 | 0.409 | 0.232 | 0.171 | 0.223 | 0.284 | 0.078 | 0.074 | 0.082 | 0.128 |
| RotatE (std) | ± 0.012 | ± 0.007 | ± 0.013 | ±0.011 | ± 0.007 | ± 0.010 | ± 0.007 | ±0.010 | ± 0.006 | ± 0.007 | ± 0.009 | ±0.011 |
| RE-NET | 0.261 | 0.210 | 0.298 | 0.410 | 0.261 | 0.210 | 0.251 | 0.331 | 0.232 | 0.139 | 0.241 | 0.369 |
| RE-NET (std) | ± 0.010 | ± 0.009 | ± 0.009 | ±0.012 | ± 0.006 | ± 0.009 | ± 0.011 | ±0.010 | ± 0.012 | ± 0.009 | ± 0.012 | ±0.018 |
| LAN | 0.230 | 0.154 | 0.247 | 0.352 | 0.185 | 0.133 | 0.201 | 0.287 | 0.207 | 0.119 | 0.234 | 0.321 |
| LAN (std) | ± 0.010 | ± 0.007 | ± 0.009 | ±0.012 | ± 0.015 | ± 0.014 | ± 0.011 | ±0.007 | ± 0.009 | ± 0.014 | ± 0.015 | ±0.013 |
| I-GEN | 0.303 | 0.238 | 0.323 | 0.420 | 0.221 | 0.179 | 0.229 | 0.264 | 0.212 | 0.120 | 0.251 | 0.346 |
| I-GEN (std) | ± 0.013 | ± 0.011 | ± 0.011 | ±0.015 | ± 0.017 | ± 0.013 | ± 0.016 | ±0.020 | ± 0.014 | ± 0.012 | ± 0.019 | ±0.012 |
| T-GEN | 0.292 | 0.218 | 0.310 | 0.394 | 0.234 | 0.185 | 0.222 | 0.271 | 0.169 | 0.122 | 0.184 | 0.265 |
| T-GEN (std) | ± 0.027 | ± 0.024 | ± 0.028 | ±0.041 | ± 0.028 | ± 0.021 | ± 0.030 | ±0.033 | ± 0.016 | ± 0.013 | ± 0.020 | ±0.022 |
| MetaDyGNN | 0.350 | 0.270 | 0.379 | 0.511 | 0.309 | 0.238 | 0.309 | 0.459 | 0.307 | 0.216 | 0.309 | 0.469 |
| MetaDyGNN (std) | ± 0.013 | ± 0.009 | ± 0.011 | ±0.014 | ± 0.008 | ± 0.006 | ± 0.011 | ±0.013 | ± 0.006 | ± 0.004 | ± 0.007 | ±0.011 |
| MetaTKGR | 0.370* | 0.303* | 0.416* | 0.558* | 0.329* | 0.253* | 0.335* | 0.489* | 0.335* | 0.249* | 0.340* | 0.527* |
| MetaTKGR (std) | ± 0.016 | ± 0.012 | ± 0.015 | ±0.008 | ± 0.012 | ± 0.014 | ± 0.012 | ±0.014 | ± 0.009 | ± 0.015 | ± 0.014 | ±0.010 |

## A.5 Detailed Experimental Results

In this section, we report the complete experimental results of 1/2/3-shot temporal knowledge graph reasoning tasks in Table 1, 2, 3, measured by $MRR$ and $Hit@\{1, 3, 10\}$. Average results on 5 independent runs with different random seeds are reported. $*$ indicates the statistically significant improvements over the best baseline, with $p$-value smaller than 0.001. We also report the detailed standard deviation for all models. MetaTKGR can beat all state-of-the-art baselines on all datasets, and we observe that MetaTKGR and most baselines are stable w.r.t. the performance with small standard deviations.