# OpenReview forum: "Learning to Sample and Aggregate: Few-shot Reasoning over Temporal Knowledge Graphs"
_NeurIPS.cc/2022/Conference — NeurIPS 2022 Accept_

### Official Review · Reviewer_oR3L · 2022-07-10

**Rating:** 7
**Confidence:** 3
**Soundness:** 3 good
**Presentation:** 3 good
**Contribution:** 3 good

**Summary:**

This paper aims to predict future facts about *newly emerging entities in temporal knowledge graphs (TKGs)* using few-shot learning. The authors demonstrated the challenge of predicting facts in temporal knowledge graphs as opposed to static knowledge graphs; temporal graphs evolve over time, making reasoning tasks (such as linking prediction) more complex. These entities pose several challenges, such as few-shot relationships and time-shifting, where characteristics may change over time. Furthermore, the previous works address this task by ignoring the emergence of new entities over time and randomly simulating unseen entities on static knowledge graphs.
In this regard, the authors propose **MetaTKGR**, a meta-learning framework for reasoning over temporal knowledge graphs. Here, the meta-learning technique is employed to dynamically learn the strategies for sampling and aggregating neighbours in TKGs.
The authors conducted experiments on three public datasets to verify the significance of their approach. The main contributions of this paper are two-fold: I) the authors presented a formal formulation for few-shot temporal graph reasoning. II) The authors proposed a meta-learning framework for solving this problem. In the following, I list my detailed comments on the points raised in the paper.

**Questions:**

- There is no clear indication if the future facts to be predicted are predefined based on the entity type?
- How the quadruples of entities are split: into support and query sets randomly? In which ratio, Is it a hyper-parameter?
- Line 284, it's unclear how the proposed approach outperforms all baselines with the same ratio of 11.4% (relative). Given that the baselines are different in structure and methodology, it's expected to perform differently and show different results.
- Line 271, It's unclear why the authors split the time duration into four periods. Does this need to be clarified? Is it suitable for all datasets?
* The approach presented in [1] is comparable to this paper and employed the same datasets in the experiments. It makes sense to benchmark the performance against it also as a baseline.

* Line 353 and 354, the authors discuss some related works designed for reasoning tasks for static/temporal homogeneous graphs. However, most of these approaches are designed for few-shot learning on a static knowledge graph. Some recent methods [1,2,3] have been developed for temporal knowledge graphs reasoning and are not discussed in the related work section to give readers an overview of other techniques.

[1] https://dl.acm.org/doi/abs/10.1145/3404835.3462963

[2] https://dl.acm.org/doi/pdf/10.1145/3488560.3498451

[3] https://ieeexplore.ieee.org/abstract/document/9712786


**Limitations:**

Yes


**Strengths And Weaknesses:**

* The paper is well-written, and almost parts are easy to follow. The authors described the research problem in detail and clarified the motivation behind their approach.
* The problem statement is formally defined with mathematical explanations. The authors formulate clearly the problem of fact reasoning of new entities in temporal knowledge graphs, given challenges: few known facts and time-shift, which makes it easier for the readers to understand it very well.
* This paper presents an interesting research problem, specifically for time-sensitive applications, and is relevant to the NeurIPS conference.
* The authors conducted several experiments on different datasets, including different time periods:
   * A detailed description of the experimental setting is provided to reproduce the results.
   * The experimental results demonstrated the significance of the proposed approach. The results are reported on five independent runs, with a significance test p-value < 0.001.
   * Further, the authors analysed the experimental results from a theoretical perspective.

---

> ### Author Response · Authors · 2022-08-02
> **Reply to Reviewer oR3L (1/2)**
>
> Thank you for the detailed comments and valuable questions. Next, we provide details to clarify your questions.
>
> > Q1: There is no clear indication if the future facts to be predicted are predefined based on the entity type?
>
> In short, we do not either predefine future facts based on the entity type or utilize any entity type information. In this paper, we utilize a quadruple to represent a fact $(s, r, o, t)$ (subject, relation, object, time), and by future facts, we refer to those happening after a split time $T$, i.e., {$(s_i, r_i, o_i, t_i) | t_i > T$}. Therefore, we define future facts by split time $T$ rather than the entity type. To predict future facts for each new entity, the only information we know ahead of time is (new entity, relation, time), based on which we aim to predict missing entity: $(\tilde{e}, r_i, ?, t)$ or $(?, r_i, \tilde{e}, t)$. We do not utilize any entity type information.
>
> Our results do not rely on entity type information. And if datasets provide the entity type information, it can be easily adopted into MetaTKGR by adding specific links into graphs.
>
> > Q2: How the quadruples of entities are split: into support and query sets randomly? In which ratio, Is it a hyper-parameter?
>
> * For each new entity, support and query sets are splitted in term of timestamps, not randomly;
> * In our $K$-shot experiments, only the first $K$ quadruples of each new entity are viewed as support sets, and the remaining quadruples are collected as query sets. During the test phase, we assume only the support sets are available and aim to predict all quadruples in corresponding query sets.
> * $K$ is a hyper-parameter, and we vary it from $1$ to $3$ to conduct main experiments shown in Table 2&3 and Figure 3&4, which demonstrates our model can generally achieve the best performance in extreme few-shot settings. And in the cross-shot experiment, we vary $K$ from $1$ to $5$ randomly to demonstrate MetaTKGR's robustness against varied number of observable quadruples  (as shown in Table 4).
>
> > Q3: Line 284, it's unclear how the proposed approach outperforms all baselines with the same ratio of 11.4\% (relative). Given that the baselines are different in structure and methodology, it's expected to perform differently and show different results.
>
> Thanks for pointing it out. We want to kindly clarify that those relative improvement ratios are calculated over **the best baseline** per each metric and per each $K$-shot experiment. Then we calculate the average relative improvement ratio: 11.4\%. We rephrased it in the modified paper.
>
> > Q4: Line 271, It's unclear why the authors split the time duration into four periods. Does this need to be clarified? Is it suitable for all datasets?
>
> Good question! The major reason for why we choose to split the time duration into four periods (background/meta-training/meta-validation/meta-test) instead of utilizing standard three-period scheme (meta-training/meta-validation/meta-test) is to better simulate real scenarios. In the real temporal knowledge reasoning cases, people can train their models on the existing KGs initially, and they need to adapt their models to new entities as time goes by.
>
> In our split scheme, we first simulate the existing KGs by background data, which can be utilized to train the model parameters initially. Then we simulate the new entities in meta-training period to enable bi-level optimization in MetaTKGR, which can make the models parameters to be easily adapted to new entities over time with few-shot link. Finally,  we use true new-appearing entities in meta-validation/meta-test period to validate/test the performance.
>
> Also, we want to point out that such split scheme can be applied to all temporal knowledge graph datasets. We need three user-defined time/ratio to split into four parts, while conventional split scheme requires two. The only extra requirement is one user-defined time to split background periods.

---

> > ### Author Response · Authors · 2022-08-02
> > **Reply to Reviewer oR3L (2/2)**
> >
> > > Q5: The approach presented in [1] is comparable to this paper and employed the same datasets in the experiments. It makes sense to benchmark the performance against it also as a baseline.
> >
> > Thanks for the suggestion. We evaluate this baseline model and report the performance in the following table as well as in modified Appendix D.
> >
> > | YAGO  | 3-shot  |||| 2-shot |||| 1-shot ||||
> > |:-------------:|:---------------:|:-------------:|:---------------:|:-------------:|:---------------:|:-------------:|:---------------:|:-------------:|:---------------:|:-------------:|:---------------:|:---------------:|
> > Metrics | MRR | H@1 | H@3 | H@10 | MRR | H@1 | H@3 | H@10 | MRR | H@1 | H@3 | H@10 | MRR | H@1 | H@3 | H@10 |
> > | RE-GCN | 0.283 | 0.226 | 0.307 | 0.421 | 0.241 | 0.225 | 0.280 | 0.407 | 0.233 | 0.161 | 0.237 | 0.320 |
> > | MetaTKGR  | 0.370 | 0.303 | 0.416 | 0.558 | 0.356 | 0.284 | 0.372 | 0.526 | 0.294 | 0.240 | 0.319 | 0.428 |
> >
> > | WIKI  | 3-shot  |||| 2-shot |||| 1-shot ||||
> > |:-------------:|:---------------:|:-------------:|:---------------:|:-------------:|:---------------:|:-------------:|:---------------:|:-------------:|:---------------:|:-------------:|:---------------:|:---------------:|
> > Metrics | MRR | H@1 | H@3 | H@10 | MRR | H@1 | H@3 | H@10 | MRR | H@1 | H@3 | H@10 | MRR | H@1 | H@3 | H@10 |
> > | RE-GCN | 0.277 | 0.223 | 0.262 | 0.344 | 0.247 | 0.202 | 0.249 | 0.310 | 0.223 | 0.183 | 0.179 | 0.250 |
> > | MetaTKGR | 0.329 | 0.253 | 0.335 | 0.489 | 0.309 | 0.249 | 0.325 | 0.441 | 0.277 | 0.203 | 0.307 | 0.419 |
> >
> > | ICEWS18  | 3-shot  |||| 2-shot |||| 1-shot ||||
> > |:-------------:|:---------------:|:-------------:|:---------------:|:-------------:|:---------------:|:-------------:|:---------------:|:-------------:|:---------------:|:-------------:|:---------------:|:---------------:|
> > Metrics | MRR | H@1 | H@3 | H@10 | MRR | H@1 | H@3 | H@10 | MRR | H@1 | H@3 | H@10 | MRR | H@1 | H@3 | H@10 |
> > | RE-GCN  | 0.233 | 0.142 | 0.238 | 0.350 | 0.205 | 0.117 | 0.229 | 0.347 | 0.193 | 0.135 | 0.218 | 0.247 |
> > | MetaTKGR    | 0.335 | 0.249 | 0.340 | 0.527 | 0.300 | 0.208 | 0.311 | 0.500 | 0.295 | 0.207 | 0.309 | 0.496 |
> >
> > As RE-GCN is not specifically designed for predictions of new entities, we train it with background set, meta-training set as well as first $K$ quadruples of each new entity in meta-validation and meta-test sets. We choose embedding dimension as 128 for fair comparison, and utilize other important hyper-parameters provided by [1]. Following [1], We remove the static graph constraint on WIKI, YAGO, as there lacks the required static information in these datasets.
> >
> > Generally speaking, RE-GCN achieves better performance than another temporal knowledge graph reasoning baseline RE-NET, and the improvements are more significant on YAGO and WIKI datasets. We hypothesize that it is because YAGO and WIKI have longer time interval, providing more stable information. MetaTKGR can beat RE-GCN on three datasets by a large margin. Compared with RE-GCN, MetaTKGR can address few-shot challenge by the proposed temporal meta-learning framework. We will add RE-GCN and the discussion in our experiment section in the new version.
> >
> > > Q6: Line 353 and 354, the authors discuss some related works designed for reasoning tasks for static/temporal homogeneous graphs. However, most of these approaches are designed for few-shot learning on a static knowledge graph. Some recent methods [1,2,3] have been developed for temporal knowledge graphs reasoning and are not discussed in the related work section to give readers an overview of other techniques.
> >
> > Thanks for the very constructive suggestion. We will add the discussion of these recent methods in Temporal Knowledge Graph Reasoning part in Section 5. Currently, due to page limitation, we detailedly compare and discuss them in Appendix E.
> >
> > [1] Li, Zixuan and Jin, Xiaolong and Li, Wei and Guan, Saiping and Guo, Jiafeng and Shen, Huawei and Wang, Yuanzhuo and Cheng, Xueqi. Temporal Knowledge Graph Reasoning Based on Evolutional Representation Learning.
> >
> > [2] Namyong Park, Fuchen Liu, Purvanshi Mehta, Dana Cristofor, Christos Faloutsos, Yuxiao Dong. EvoKG: Jointly Modeling Event Time and Network Structure for Reasoning over Temporal Knowledge Graphs.
> >
> > [3] Geng, Yipeng and Shao, Yali and Zhang, Shanwen and He, Xiaoyun. Multi-hop Temporal Knowledge Graph Reasoning over Few-Shot Relations with Novel Method.

---

> > > ### Author Response · Authors · 2022-08-08
> > > **Thank You and Look Forward to Your Reply**
> > >
> > > We sincerely thank the reviewer oR3L for providing valuable comments to improve our paper. Hopefully, you had a chance to go over our responses. Please let us know if we have addressed all your concerns and questions or not. We look forward to a fruitful discussion and your replies, which are highly valuable to us.

---

### Official Review · Reviewer_KWLf · 2022-07-11

**Rating:** 5
**Confidence:** 3
**Soundness:** 3 good
**Presentation:** 3 good
**Contribution:** 3 good

**Summary:**

This paper investigated a novel problem, few-shot temporal knowledge graph reasoning, that aims to predict future facts for newly emerging entities based on limited observations in evolving graphs. To address this problem, the paper proposed a novel framework, namely MetaTKGR, which aims to extract the global knowledge of sampling and aggregating temporal neighbors, which can be adapted to new entities for future prediction. The experiment results on three real-world temporal knowledge graphs are promising.

**Questions:**

1. Where is the error bar in Section 4 of the main paper?

**Limitations:**

No.

**Strengths And Weaknesses:**

Strengths:
1. The paper is well-written and easy to follow.
2. The motivation is clear and interesting.
3. The experimental results and discussion are thorough.

Weaknesses:
1. It is a little hard to understand how the MetaTKGR can address the two challenges, few-shot and time shift. Maybe adding some specific examples for visualization is helpful.

---

> ### Author Response · Authors · 2022-08-02
> **Reply to Reviewer KMLf**
>
> Thank you for the comments and valuable questions. Next we provide details to clarify your major concerns.
>
> > Q1: How does MetaTKGR address few-shot and time-shift challenges?
>
> Generally speaking, to address the few-shot and time shift challenges, we expect the model parameters to be easily adapted to new entities with limited links and to maintain temporal robustness over time. This goal is achieved by a bi-level optimization (inner optimization and outer optimization). Specifically:
> * MetaTKGR first initializes entity-specific parameters from global parameters $\phi$ (step 1);
> * Then during the **inner optimization** (step 2), it fine-tunes entity-specific parameters on $K$-shot support set in order to fully utilize few-shot links and achieve better prediction performance for new entities;
> * To improve the quality of global parameters (which is the initialization of entity-specific parameters), during the **outer optimization** (step 3), MetaTKGR optimizes $\phi$ by adapting the updated $\phi_{\tilde{e}}$ on query sets and minimizing the prediction loss.
>
> Next, we explain how MetaTKGR can address few-shot and time shift challenges in detail:
>
> (1) To address the few-shot challenge: the proposed bi-level optimization follows principles of model-agnostic meta-learning (MAML) for improving fast adaptation in the few-shot regime. During the meta-training phase, MetaTKGR can learn a set of good global parameters $\phi$ that encode general knowledge as initialization. Then during the meta-test phase, $\phi$ can be fastly adapted to new entities via a few step of updates on few-shot links. Compared with training $\phi$ from scratch on new entities, MetaTKGR requires much fewer labeled data for new entities. Compared with directly utilizing $\phi$ from existing entities to new entities, MetaTKGR can better utilizes information of new entities via the fine-tuning.
>
> (2) To address the time shift challenge:
>
> (2a) MetaTKGR quantitatively measures how well the current global parameters $\phi$ performs by the performance on predicting future facts. In the outer optimization, we gradually adapt $\phi$ from near facts to further away facts, and calculate the training signals to update $\phi$. Therefore, the global parameters $\phi$ can maintain temporal robustness, i.e., achieving stably good performance for both near facts and further away facts.
>
> (2b) As new entities are highly evolving over time, the future facts may follow different and unknown distribution. Therefore, to better estimate the training signal from future unknown distributions in the outer optimization, we propose a temporal adaptation regularizer which can provide stability and improve the generalization ability of the global parameters over time.
>
> **We provided an illustrative example in Appendix B to help reviewers to better understand our proposed MetaTKGR framework.**
>
> > Q2: Where is the error bar of Section 4?
>
> * Error bar of main experimental results (Table 2,3) are reported in Appendix A.5 (Table 1,2,3), due to page limitation;
> * Error bar of ablation studies (Figure 4) are visualized as black bold lines on top of each bar in Figure 4.
>
> To calculate such error bars, we run each model for $5$ independent times, and calculate standard deviation. Generally, most baselines, ablation variants, and MetaTKGR produce stable performance with low variation. It is also worth noting that MetaTKGR achieves statistically significant improvements over the best baseline, with p-value smaller than 0.001 (as shown in Table 2,3).

---

> > ### Author Response · Authors · 2022-08-08
> > **Thank You and Look Forward to Your Reply**
> >
> > We sincerely thank the reviewer KMLf for providing valuable comments to improve our paper. Hopefully, you had a chance to go over our responses. Please let us know if we have addressed all your concerns and questions or not. We look forward to a fruitful discussion and your replies, which are highly valuable to us.

---

### Official Review · Reviewer_NqGR · 2022-07-12

**Rating:** 6
**Confidence:** 2
**Soundness:** 4 excellent
**Presentation:** 4 excellent
**Contribution:** 3 good

**Summary:**

This paper mainly works on knowledge graph reasoning with temporal shift, which is an underexplored problem in real-world setting where the new entities will evolve over time. The challenges mainly lie in few-shot and time shift perspective. In order to solve this problem, this paper proposes a MetaTKGR framework, which can adjust the strategy of sampling and aggregating neighbors from recent facts for the new entities. The paper uses theoretical analysis to provide PAC generalization bound to prove its learnability.

The empirical results shown in the paper indicate strong performance across the board. Under 1-shot and 2/3-shot cases, the model is able to achieve consistent performance gain over different embedding-based KGR models.1

**Questions:**

N/A

**Ethics Review Area:**

["I don’t know"]

**Limitations:**

The compared baseline models are all embedding-based knowledge graph reasoning models, it's not clear whether the proposed method could be used for path-based reasoning models. The path-based reasoning models can normally provide better explainability while achieving better results.

**Strengths And Weaknesses:**

Strength:

1. This paper discovers a novel problem of knowledge graph reasoning over temporal shift.
2. This paper proposes MetaTKGR accompanied with theoretical analysis.
3. The empirical results are strong.


Weakness:

1. The propose method is a bit hard to follow, consisting a total of (5) steps indicated in Figure 2. It would be great to associate the figure with different subsections to help readers follow the logic more closely.

---

> ### Author Response · Authors · 2022-08-02
> **Reply to Reviewer NqGR (1/2)**
>
> Thank you for the comments and valuable suggestions about paper organization! We have already adjusted Figure 2 and Section 3 accordingly to make the logic more clear. We recommend Reviewer NqGR to read the updated Figure 2 and Algorithm 2 for the overview of MetaTKGR. Next we provide details to clarify your major concerns.
>
> > Q1: Overall framework of MetaTKGR:
>
> Thanks for the constructive suggestion. We have simplified Figure 2 (now we utilize **three steps** to summarize the training procedure of MetaTKGR framework), and added high-level descriptions to make logic easier to follow.
>
> Generally speaking, to address the few-shot and time shift challenges, we expect the model parameters to be easily adapted to new entities with limited links and to maintain temporal robustness over time. This goal is achieved by a bi-level optimization (inner optimization and outer optimization). Specifically:
> * MetaTKGR first initializes entity-specific parameters $\phi_{\tilde{e}}$ from global parameters $\phi$ (step 1);
> * Then during the **inner optimization** (step 2), it fine-tunes entity-specific parameters $\phi_{\tilde{e}}$ on $K$-shot support set in order to fully utilize few-shot links and achieve better prediction performance for new entities;
> * To improve the quality of global parameters (which is the initialization of entity-specific parameters), during the **outer optimization** (step 3), MetaTKGR optimizes $\phi$ by adapting the updated $\phi_{\tilde{e}}$ on query sets and minimizing the prediction loss.
>
> We polished the structure of Section 3 accordingly, which is structured as follows:
> * Section 3.2 gives an overview of MetaTKGR;
> * Section 3.3 introduces our temporal encoder (to specify $f_\phi$);
> * Line 198-212 in Section 3.4 introduces the inner optimization;
> * Line 213-252 in Section 3.4 introduces the outer optimization.
>
> Next, we explain how MetaTKGR can address few-shot and time shift challenges in detail:
>
> (1) To address the few-shot challenge: the proposed bi-level optimization follows principles of model-agnostic meta-learning (MAML) [1] to support fast adaptation in the few-shot regime. During the meta-training phase, MetaTKGR can learn a set of good global parameters $\phi$ that encode general knowledge as initialization. Then during the meta-test phase, $\phi$ can be fastly adapted to new entities via a few step of updates on few-shot links. Compared with training $\phi$ from scratch on new entities, MetaTKGR requires much fewer labeled data for new entities. Compared with directly utilizing $\phi$ from existing entities to new entities, MetaTKGR can better utilizes information of new entities via the fine-tuning.
>
> (2) To address the time shift challenge:
>
> (2a) MetaTKGR quantitatively measures how well the current global parameters $\phi$ performs by the performance on predicting future facts. In the outer optimization, we gradually adapt the model from near facts to further away facts, and calculate the training signals to update $\phi$. Therefore, the global parameters $\phi$ can maintain temporal robustness, i.e., achieving stably good performance for both near facts and further away facts.
>
> (2b) As new entities are highly evolving over time, the future facts may follow different and unknown distribution. Therefore, to better estimate the training signal from future unknown distributions in the outer optimization, we propose a temporal adaptation regularizer which can provide stability and improve the generalization ability of the global parameters over time.
>
> > Q2: The compared baseline models are all embedding-based knowledge graph reasoning models, it's not clear whether the proposed method could be used for path-based reasoning models. The path-based reasoning models can normally provide better explainability while achieving better results:
>
> Good question! First of all, we want to highlight that our proposed temporal meta-learning framework is a model-agnostic and gradient-based method. Therefore, our bi-level optimization can be generally applied to both embedding-based and path-based (rule-based) knowledge graph reasoning methods that can be learned via the gradient-based optimizers. Conventional path-based methods are good at logical inference and have good interpretability, but they have difficulties to handle noisy data and take advantage of gradient-based optimizations to update rules. Some recent works combine neural reasoning and symbolic reasoning to address the problems, e.g., KALE [2], pLogicNet [3], DeepPath [4].

---

> > ### Author Response · Authors · 2022-08-02
> > **Reply to Reviewer NqGR (2/2)**
> >
> > It is a promising direction to extend our framework to those recent path-based (rule-based) methods, which facilitates the fast adaptation for new entities over time and meanwhile improves the interpretibility. To verify our meta temporal reasoning algorithm can be also applied to the path-based methods, we add one additional experiment to compare one path-based method KALE and the modified version with our meta temporal reasoning algorithm MetaKALE. We report the performance in the following table:
> >
> > | YAGO  | 3-shot  |||| 2-shot |||| 1-shot ||||
> > |:-------------:|:---------------:|:-------------:|:---------------:|:-------------:|:---------------:|:-------------:|:---------------:|:-------------:|:---------------:|:-------------:|:---------------:|:---------------:|
> > Metrics | MRR | H@1 | H@3 | H@10 | MRR | H@1 | H@3 | H@10 | MRR | H@1 | H@3 | H@10 | MRR | H@1 | H@3 | H@10 |
> > | KALE   | 0.231 | 0.166  | 0.250 | 0.367 | 0.198 | 0.144 | 0.217 | 0.319 | 0.192 | 0.142 | 0.209 | 0.277 |
> > | MetaKALE  | 0.241 | 0.180 | 0.277 | 0.388 | 0.223 | 0.156 | 0.231 | 0.331 | 0.216 | 0.153 | 0.216 | 0.292 |
> > | Gain (%) | 4.3 | 8.4 | 10.8 | 5.7 | 12.6 | 8.3 | 6.4 | 3.8 | 12.5 | 7.7 | 3.3 | 5.4 |
> >
> > | WIKI  | 3-shot  |||| 2-shot |||| 1-shot ||||
> > |:-------------:|:---------------:|:-------------:|:---------------:|:-------------:|:---------------:|:-------------:|:---------------:|:-------------:|:---------------:|:-------------:|:---------------:|:---------------:|
> > Metrics | MRR | H@1 | H@3 | H@10 | MRR | H@1 | H@3 | H@10 | MRR | H@1 | H@3 | H@10 | MRR | H@1 | H@3 | H@10 |
> > | KALE  | 0.173 | 0.120 | 0.185 | 0.250 | 0.155 | 0.131 | 0.143 | 0.214 | 0.147 | 0.121 | 0.145 | 0.197 |
> > | MetaKALE   | 0.180 | 0.132 | 0.199 | 0.275 | 0.173 | 0.146 | 0.160 | 0.230 | 0.169 | 0.130 | 0.158 | 0.211 |
> > | Gain (%) | 4.0 | 10.0 | 7.6 | 10.0 | 11.6 | 11.5 | 11.9 | 7.5 | 15.0 | 7.4 | 9.0 | 7.1|
> >
> > | ICEWS18  | 3-shot  |||| 2-shot |||| 1-shot ||||
> > |:-------------:|:---------------:|:-------------:|:---------------:|:-------------:|:---------------:|:-------------:|:---------------:|:-------------:|:---------------:|:-------------:|:---------------:|:---------------:|
> > Metrics | MRR | H@1 | H@3 | H@10 | MRR | H@1 | H@3 | H@10 | MRR | H@1 | H@3 | H@10 | MRR | H@1 | H@3 | H@10 |
> > | KALE | 0.073 | 0.063 | 0.088 | 0.108 | 0.071 | 0.059 | 0.089 | 0.110 | 0.061 | 0.059 | 0.071 | 0.093 |
> > | MetaKALE    | 0.091 | 0.082 | 0.099 | 0.120 | 0.088 | 0.071 | 0.100 | 0.132 | 0.083 | 0.073 | 0.089 | 0.107 |
> > | Gain (%) | 24.7 | 30.2 | 12.5 | 11.1 | 23.9 | 20.3 | 12.4 | 20.0 | 36.0 | 23.7 | 25.3 | 15.0 |
> >
> > We train the KALE with background set (initial training set for training the temporal encoder), meta-training set as well as first K triples of each new entity in meta-validation and meta-test sets. Following [2], we first utilize TransE to generate rules, and apply KALE to learn both triplets and rules. To enable meta-training of MetaKALE in our framework, we view the embeddings as model parameters $\phi$, utilize temporal information to sort triplets, and meta-train global $\phi$ similar to Algorithm 2. We observe that KALE can slightly beat TransE model, and MetaKALE further improves KALE by 12.7% on average for new entities. This demonstrates that our framework is general and compatible to path-based (rule-based) knowledge graph reasoning baselines.
> >
> > [1] Chelsea Finn, Pieter Abbeel, and Sergey Levine. Model-agnostic meta-learning for fast adap416 tation of deep networks.
> >
> > [2] S. Guo, Q. Wang, L. Wang, B. Wang, and L. Guo. Jointly embedding knowledge graphs and logical rules.
> >
> > [3] M. Qu and J. Tang. Probabilistic logic neural networks for reasoning.
> >
> > [4] W. Xiong, T. Hoang, and W. Y. Wang. Deeppath: A reinforcement learning method for knowledge graph reasoning.

---

> > > ### Author Response · Authors · 2022-08-08
> > > **Thank You and Look Forward to Your Reply**
> > >
> > > We sincerely thank the reviewer NqGR for providing valuable comments to improve our paper. Hopefully, you had a chance to go over our responses. Please let us know if we have addressed all your concerns and questions or not. We look forward to a fruitful discussion and your replies, which are highly valuable to us.

---

### Author Response · Authors · 2022-08-02
**Reply to all: summary of the major revision**

We sincerely thank all reviewers for the thorough and detailed reviews on our submission. The comments “This paper is well-written, and almost parts are easy to follow”, “This paper presents an interesting problem”, "The empirical results are strong" are very encouraging! We summarize major changes that we have made below. All changes are marked in blue in the updated paper and supplementary material.

* We modified the description of Figure 2 to help reviewers better understand the proposed MetaTKGR framework, and associated Figure 2a to Definition 2.3 (task illustration), Figure 2b to Section 3.3 (the temporal encoder), Figure 2c to Section 3.4 (the bi-level optimization); (to address Reviewer NqGR's concern)
* We also provided an illustrative example (1-shot learning on one example entity) in Appendix B to facilitate reviewers to better understand our proposed MetaTKGR framework; (corresponding to Reviewer KMlF's suggestion)
* We added one additional experiment in Appendix C to validate our MetaTKGR framework is general and compatible to path-based (rule-based) knowledge graph reasoning baselines, e.g., KALE [1], which brings 12.7% relative gain with the proposed meta temporal reasoning algorithm; (to address Reviewer NqGR's question)
* We added and compared one temporal baseline RE-GCN [2] in Appendix D; (based on Reviewer oR3L's comments)
* We polished related work section to discuss and compare several recent temporal knowledge graph reasoning models (i.e., RE-NET [2], EvoKG [3], FS-Path [4]) in Appendix E; (based on Reviewer oR3L's comments)

[1] S. Guo, Q. Wang, L. Wang, B. Wang, and L. Guo. Jointly embedding knowledge graphs and logical rules.

[2] Li, Zixuan and Jin, Xiaolong and Li, Wei and Guan, Saiping and Guo, Jiafeng and Shen, Huawei and Wang, Yuanzhuo and Cheng, Xueqi. Temporal Knowledge Graph Reasoning Based on Evolutional Representation Learning.

[3] Namyong Park, Fuchen Liu, Purvanshi Mehta, Dana Cristofor, Christos Faloutsos, Yuxiao Dong. EvoKG: Jointly Modeling Event Time and Network Structure for Reasoning over Temporal Knowledge Graphs.

[4] Geng, Yipeng and Shao, Yali and Zhang, Shanwen and He, Xiaoyun. Multi-hop Temporal Knowledge Graph Reasoning over Few-Shot Relations with Novel Method.

---

> ### Author Response · Authors · 2022-08-07
> **Thank You and Look Forward to Your Reply**
>
> We sincerely thank the reviewers for providing valuable comments to improve our paper. Hopefully, you had a chance to go over our responses. Please let us know if we have addressed all your concerns and questions or not. We look forward to a fruitful discussion and your replies, which are hignly valuable to us. We would appreciate it if you consider increasing your scores if our response can address your questions.

---

### Meta-Review · Area_Chair_1bHz · 2022-08-30

**Recommendation:** Accept
**Confidence:** Certain

**Metareview:**

This paper studies few-shot temporal knowledge graph reasoning. The task looks to predict future facts for newly emerging entities based on limited observations in evolving graphs. The authors propose MetaTKGR, a meta-learning framework for reasoning over temporal knowledge graphs.

The reviewers agree that the proposed method is reasonable and sound, the experiments are thorough, and the results provide valuable insights for future work. Reviewers' raised concerns and questions are properly addressed by the author's response.

**Award:**

No

---

### Decision · Program_Chairs · 2022-09-14

Accept